# Structures of NF-κB p52 homodimer-DNA complexes rationalize binding mechanisms and transcription activation

**Wenfei Pan[1†], Vladimir A Meshcheryakov[1†‡], Tianjie Li[2†], Yi Wang[2], Gourisankar Ghosh[3], Vivien Ya-Fan Wang[1,4,5]\***

[1]Faculty of Health Sciences, University of Macau, Taipa, China; [2]Department of Physics, Chinese University of Hong Kong, Shatin, Hong Kong; [3]Department of Chemistry and Biochemistry, University of California, San Diego, La Jolla, United States; [4]MoE Frontiers Science Center for Precision Oncology, University of Macau, Taipa, Macao; [5]Cancer Centre, Faculty of Health Sciences, University of Macau, Taipa, China

**\*For correspondence:** vivienwang@um.edu.mo

[†]These authors contributed equally to this work

**Present address:** [‡]Molecular Cryo-Electron Microscopy Unit, Okinawa Institute of Science and Technology Graduated University, Okinawa, Japan

**Competing interest:** The authors declare that no competing interests exist.

**Abstract** The mammalian NF-κB p52:p52 homodimer together with its cofactor Bcl3 activates transcription of κB sites with a central G/C base pair (bp), while it is inactive toward κB sites with a central A/T bp. To understand the molecular basis for this unique property of p52, we have determined the crystal structures of recombinant human p52 protein in complex with a P-selectin(PSel)-κB DNA (5′-GGGGT**G**ACCCC-3′) (central bp is underlined) and variants changing the central bp to A/T or swapping the flanking bp. The structures reveal a nearly two-fold widened minor groove in the central region of the DNA as compared to all other currently available NF-κB-DNA complex structures, which have a central A/T bp. Microsecond molecular dynamics (MD) simulations of free DNAs and p52 bound complexes reveal that free DNAs exhibit distinct preferred conformations, and p52:p52 homodimer induces the least amount of DNA conformational changes when bound to the more transcriptionally active natural G/C-centric PSel-κB, but adopts closed conformation when bound to the mutant A/T and swap DNAs due to their narrowed minor grooves. Our binding assays further demonstrate that the fast kinetics favored by entropy is correlated with higher transcriptional activity. Overall, our studies have revealed a novel conformation for κB DNA in complex with NF-κB and pinpoint the importance of binding kinetics, dictated by DNA conformational and dynamic states, in controlling transcriptional activation for NF-κB.

## Editor's evaluation

This manuscript provides an important structural and biophysical characterization of several complexes of the p52 homodimer of NF kB and different DNA binding sites. The main topic is the investigation of why the central base pair(s) have a strong influence on the transcriptional activity of the homodimer. The authors correlate structural changes with measurements of kinetic on and off rates to develop a model that explains the differences in activity and supports their interpretation with MD simulations, however, some technical issues regarding differences in the ITC and BLI measurements remain unresolved. The paper will be of interest to scientists working on understanding transcriptional regulation.

## Introduction

The binding of transcription factors (TFs) to their specific DNA response elements in the promoters/enhancers of target genes is the key event regulating gene transcription and consequent cellular processes. For proper gene expression, TFs must interact selectively at the correct place and time and assemble into high-order complexes with specific DNA sequences and cofactors (*Natoli et al., 2005*; *Mulero et al., 2019*). In eukaryotic genomes, the ability of TFs to select a small subset of relevant binding sites out of the large excess of potential binding sites within the genomes is the foundation upon which transcriptional regulation is built. Structural studies have provided valuable information on how various DNA binding domains recognize their cognate DNA binding sites at atomic resolution (*Garvie and Wolberger, 2001*). However, how TFs discriminate between closely related, but biologically distinct, DNA sequences are not well understood.

The nuclear factor κB (NF-κB) family of TFs regulates diverse biological responses (*Zhang et al., 2017*). Mammalian NF-κB is assembled combinatorially from five subunits, p50/NF-κB1, p52/NF-κB2, RelA/p65, c-Rel, and RelB, into homo- and heterodimers which bind to specific DNA sequences, known as κB site or κB DNA. All five subunits share a highly conserved region at their N-termini, referred to as the Rel homology region (RHR), and the three-dimensional structures of the RHR are also highly conserved among these proteins. The RHR is roughly 300 residues in length and contains the N-terminal domain (NTD), dimerization domain (DD), and nuclear localization signal (NLS). The DD alone mediates protein homo- and heterodimerization; the NTD and DD together are responsible for DNA binding; and the NLS region is flexible in solution and together with the DD forms the binding sites for the inhibitor of NF-κB (IκB) proteins.

The NF-κB proteins can be further divided into two subclasses: the p50 and p52 subunits belong to class I by virtue of their lack of a transcriptional activation domain (TAD). The other three subunits, RelA, c-Rel, and RelB, constitute class II with every member containing a TAD at its C-terminus. Mature p50 and p52 subunits are generated via incomplete proteolysis of their precursor proteins p105 and p100 (*Figure 1—figure supplement 1A*), respectively. Therefore, p50 and p52 possess a short glycine-rich region (GRR) at their C-termini.

The initial discovery and characterization of several physiological κB DNAs established the pseudo-symmetric consensus sequence as $5'\text{-}G_{-5}G_{-4}G_{-3}R_{-2}N_{-1}W_0Y_{+1}Y_{+2}C_{+3}C_{+4}\text{-}3'$ (*Lenardo and Baltimore, 1989*), where R=purines, N=any nucleotides, W=either A or T, and Y=pyrimidines. The subsequent identification of new NF-κB-DNA binding sites broadened the consensus to $5'\text{-}G_{-5}G_{-4}G_{-3}N_{-2}N_{-1}N_0N_{+1}N_{+2}C_{+3}C_{+4}\text{-}3'$ (*Chen and Ghosh, 1999*; *Mulero et al., 2019*). The critical features of the consensus κB DNA sequence are the presence of a series of G and C bases at the 5′ and 3′ ends, respectively, while the bases at the central region can vary. X-ray structures of various NF-κB dimers in complex with different κB DNAs revealed conserved protein-DNA recognition modes for κB DNA that follows the consensus sequence (*Müller et al., 1995*; *Ghosh et al., 1995*; *Cramer et al., 1997*; *Chen et al., 1998b*; *Huang et al., 2001*; *Moorthy et al., 2007*; *Fusco et al., 2009*; *Chen et al., 1998a*; *Chen et al., 2000*; *Escalante et al., 2002*; *Berkowitz et al., 2002*; *Chen-Park et al., 2002*; *Panne et al., 2007*). The RHR of each monomer binds to half of a κB DNA, called half-site. A set of conserved amino acid (aa) residues mediate base-specific contacts to the 5′ and 3′ flanking G and C bases; the inner, more variable bases participate in important, but less base-specific interactions. The length of κB DNAs varies from 9 to 11 base pairs (bps). The p50 and p52 subunits bind to a 5 bp half-site (5′-GGGNN-3′), while the RelA and c-Rel subunits prefer a 4 bp half-site (5′-NNCC-3′). Following this binding pattern, both the p50:p50 and p52:p52 homodimers prefer an 11 bp κB site comprising of two 5 bp half-sites separated by a central bp (5+1+5 bps), such as the IL-6-κB site (5′-GGGAT̲TTCC-CC-3′). On the other hand, RelA:RelA and c-Rel:c-Rel homodimers bind 9 bp κB sites containing two 4 bp half-sites (4+1+4 bps), such as the IL-8-κB site (5′-GGAA̲TTTCC-3′). Heterodimers containing one p50 or p52 subunit, such as p50:RelA and p52:RelB, recognize a 10 bp κB site (5+1+4 bps), that is, the HIV-κB site (5′-GGGAC̲TTTCC-3′) and IFN-β-κB site (5′-GGGAA̲ATTCC-3′), in which the central bp lies at the pseudo-dyad axis of the dimer and is not directly contacted by the protein.

Genome-wide NF-κB-DNA motif identification studies revealed that NF-κB associates not only with consensus κB DNAs, but also with sequences containing only one half-site consensus, and even some sequences with no consensus (*Lim et al., 2007*; *Martone et al., 2003*; *Zhao et al., 2014*). In vitro binding experiments have been carried out to classify κB DNAs according to their binding specificity for different NF-κB dimers. The binding affinity displayed by various NF-κB dimers for distinct

κB DNAs does not necessarily correlate with what occurs during regulation of gene expression in vivo. For example, the p50:RelA heterodimer binds tightly to most κB DNAs, whereas RelA:RelA and c-Rel:c-Rel homodimers bind many of the same sequences with relatively low affinity. However, detailed genetic experiments have shown that some genes are activated only in the presence of one or a subset of NF-κB subunits, such as mice lacking c-Rel exhibit defects in IL-2 and IL-12 expression (*Köntgen et al., 1995*; *Hoffmann et al., 2003*). In addition to specific gene activation, NF-κB dimers are also known to repress transcription. The RelA and p50 dimers have been shown to repress the expressions of *nrp1* gene, which is essential for osteoclast differentiation, and the interferon-stimulated response element, respectively (*Cheng et al., 2011*; *Hayashi et al., 2012*). Both of these sites also display only half-site similarity to the κB DNA consensus. Structural and biochemical analyses of NF-κB-DNA binding have also revealed the existence of a large number of κB DNAs that display relatively similar affinities compared with κB consensus even though they lack one consensus half-site entirely (*Ghosh et al., 2012*; *Siggers et al., 2011*). Therefore, in vitro data do not fully capture the complexity of DNA recognition and gene regulation by NF-κB in cells.

NF-κB p52 is generated from the precursor protein p100 (*Figure 1—figure supplement 1A*), a tightly regulated process that requires specific stimuli. Unregulated p100 processing into p52 results in multiple myeloma and other lymphoid malignancies, which is detrimental to normal cellular function (*Courtois and Gilmore, 2006*; *Annunziata et al., 2007*; *Keats et al., 2007*). We previously demonstrated that the p52:p52 homodimer could sense a single bp change from G/C to A/T at the central position of a κB DNA (*Wang et al., 2012*). The p52:p52 homodimer binds both κB DNAs; but only in the case of the G/C-centric DNA, p52:p52 homodimer can associate with its specific cofactor Bcl3 (p52:p52:Bcl3 complex) and activate transcription by recruiting histone acetyltransferases. When bound to the A/T-centric DNA, the same p52:p52:Bcl3 complex represses gene transcription through the recruitment of histone deacetylases. It is intriguing that the identity of a non-contacted nucleotide should have such a drastic effect on transcriptional selectivity. Leung et al. reported that the transcriptional activity of the RelA:RelA homodimer upon binding to the IP-10-κB DNA (5′-GGGAAATTCC-3′) and the MCP-1-κB DNA (5′-GGGAATTTCC-3′) are different; a single bp difference between the A-centric IP-10-κB and T-centric MCP-1-κB DNAs alters the genes' responsiveness to RelA (*Leung et al., 2004*). Taken together, these reports strongly suggest that NF-κB transcriptional outcomes are coded within specific κB DNA sequences. Even small changes in the promoter-specific κB DNAs, which do not alter the overall NF-κB binding affinity, might alter the gene expression profiles. Although structural studies have revealed a stereochemical mechanism of how NF-κB dimers bind κB DNAs, the effect of DNA conformation on complex formation remains underappreciated and it requires solid understanding of both structure and dynamics of κB DNAs to elucidate such a mechanism.

In the present study, we determined the crystal structures of the p52:p52 homodimer in complex with the natural G/C-centric PSel-κB DNA (5′-GGGGTGACCCC-3′) and two related DNAs where the central three positions were varied, named as PSel (mutant A/T-centric) (5′-GGGGTAACCCC-3′) and PSel (−1/+1 swap) (5′-GGGGAGTCCCC-3′). PSel is a known target gene regulated by the p52:p52:Bcl3 complex in cells (*Pan and McEver, 1995*; *Wang et al., 2012*); however, the PSel (mutant A/T-centric) and (−1/+1 swap) κB DNAs reduced the transcriptional activity significantly. All three structures revealed a widening of the DNA minor groove in the central region compared to all previously known structures of NF-κB-(A/T-centric)-DNA complexes. MD simulations showed free DNAs exist in distinct preferred conformations, and the p52:p52 homodimer induces the least amount of conformational changes on the more transcriptionally active Psel (natural G/C-centric) κB DNA which has an intrinsically widened minor groove. In vitro experiments further demonstrated that the binding kinetics, rather than the binding affinity, is correlated with their transcriptional activities. The combination of structural, MD simulations, and biochemical studies presented here provides new insights into allosteric control by closely related κB DNAs on NF-κB-dependent transcriptional specificity.

## Results

### The central base pairs in PSel-κB DNA regulate p52:p52:Bcl3 transcriptional activity

Structures of several NF-κB dimers in complex with various κB DNAs have been reported over the past 25 years. In all these structures, the DNA sequences contain A/T-centric κB sites (*Figure 1—figure*

*supplement 2E*; *Ghosh et al., 1995*; *Müller et al., 1995*; *Cramer et al., 1997*; *Chen et al., 1998a*; *Huang et al., 2001*; *Moorthy et al., 2007*; *Fusco et al., 2009*; *Chen et al., 1998b*). The PSel-κB DNA (5′-GGGGTGACCCC-3′) (the central bp is in red color, bps at ±1 positions are underlined), a natural binding site known to be specifically regulated by the p52:p52:Bcl3 complex (*Pan and McEver, 1995*; *Wang et al., 2012*), is distinctive from the canonical κB sites not only at the central position but also the two flanking positions. Whereas p50 and other subunits prefer an A:T at −1 and T:A at +1 positions, such as the MHC-κB site (5′-GGGGATTCCCC-3′), PSel-κB contains T:A and A:T at the equivalent positions, respectively. We mutated the central and flanking bps to generate PSel (mutant A/T-centric) (5′-GGGGTAACCCC-3′) and (−1/+1 swap) (5′-GGGGAGTCCCC-3′) DNAs. Transcriptional activity of the p52:p52:Bcl3 complex was measured for these three and MHC-κB sites using a luciferase reporter-based assay. The natural PSel luciferase reporter could be activated by endogenous NF-κB with co-expression of Bcl3 followed by lipopolysaccharide stimulation (*Figure 1A*). To investigate the effects of PSel (mutant A/T-centric) and (−1/+1 swap) on transcriptional activity, luciferase reporter constructs with the variants or MHC-κB site were co-transfected with p52 and Bcl3 expression plasmids. PSel (mutant A/T-centric) showed twofold reduced reporter activity, while both PSel (−1/+1 swap) and MHC-κB showed drastically reduced transcriptional activity as compared to the natural PSel-κB (*Figure 1B*; *Figure 1—figure supplement 1C*). These results suggest that the bp identity at all three positions in the central region is critical in determining transcriptional activity of the p52:p52:Bcl3 complex, which is in line with our previous study that the central bp of κB DNAs plays critical roles in transcriptional regulation (*Wang et al., 2012*).

## Widened minor groove in PSel-κB DNA in complex with NF-κB p52:p52 homodimer

Since only p52 mediates DNA interactions in the p52:p52:Bcl3 complex (*Figure 1—figure supplement 1D*; *Bours et al., 1993*), we focused our study on (p52:p52)-DNA and speculated that the observed transcriptional differences could be due to different structural features of (p52:p52)-DNA complexes. We solved the crystal structures of p52:p52 homodimer in complex with all three PSel-κB DNAs (*Figure 1C–H*; *Table 1*). The p52 protein works as a bridging factor between target DNAs and Bcl3; therefore, a recombinant p52 protein (aa 1–398) which could form complex with Bcl3 was co-crystallized with the DNAs (*Figure 1—figure supplement 1E–H*). This p52 construct contains most of the GRR region which was not included in any previous NF-κB structures (*Figure 1—figure supplement 1B*, *Figure 1—figure supplement 2E*); however, no electron density was observed for the C-terminal part (aa 330–398) in the structures. The length of PSel-κB DNAs used in the co-crystallization trials ranged from 12 to 20 bp (*Supplementary file 1*). p52 (aa 1–398) protein co-crystallized with 18 and 20 bp DNAs but diffracted to a higher resolution with the 18 bp DNA.

The overall structures of p52:p52 in complex with the natural PSel-κB DNA and two variants are similar to each other with similar buried surface area and number of hydrogen bonds (H-bonds) (*Figure 1D and H*; *Supplementary file 2*). It should be noted that all the complex structures are at ~3.0 Å resolution which places limits on the identification of some detailed interactions, including hydrogen bonds. However, compared to previously known structures of NF-κB-DNA complexes, two striking differences are observed. One is that all three PSel-κB DNAs exhibited a distinct widening of the minor groove at the two base-steps around the central position 0 (−1 to 0 and 0 to +1), with width of ~7.5 Å (*Figure 1E–G1*). In comparison, the A/T-centric κB DNAs studied earlier, κB-33 (5′-GGAAATTTCC-3′) (*Chen et al., 1998a*; *Huang et al., 2005*) and another one that we now name κB-55 (5′-GGGAATTCCC-3′) (*Moorthy et al., 2007*; *Fusco et al., 2009*), have significantly compressed minor groove in both their bound and free states as compared to an ideal B-form DNA (*Figure 1I*; *Figure 1—figure supplement 2A–B*). Compressed minor groove width is a common feature of all A/T-centric κB DNAs bound to NF-κB dimers which is remarkably different from the minor groove width of the PSel-κB DNAs seen in the present structures (*Figure 1—figure supplement 2E*).

## The widened minor groove is observed with long p52 proteins

The other difference observed for the three p52:p52 structures reported here concerns the organization of the dimer and the complex with DNA. The p52-MHC-κB DNA (which is A/T-centric) complex is the only previously determined crystal structure of NF-κB p52:p52 homodimer (*Cramer et al., 1997*). Superposition of the p52:p52 homodimer in the MHC-κB and natural PSel-κB complexes aligned by

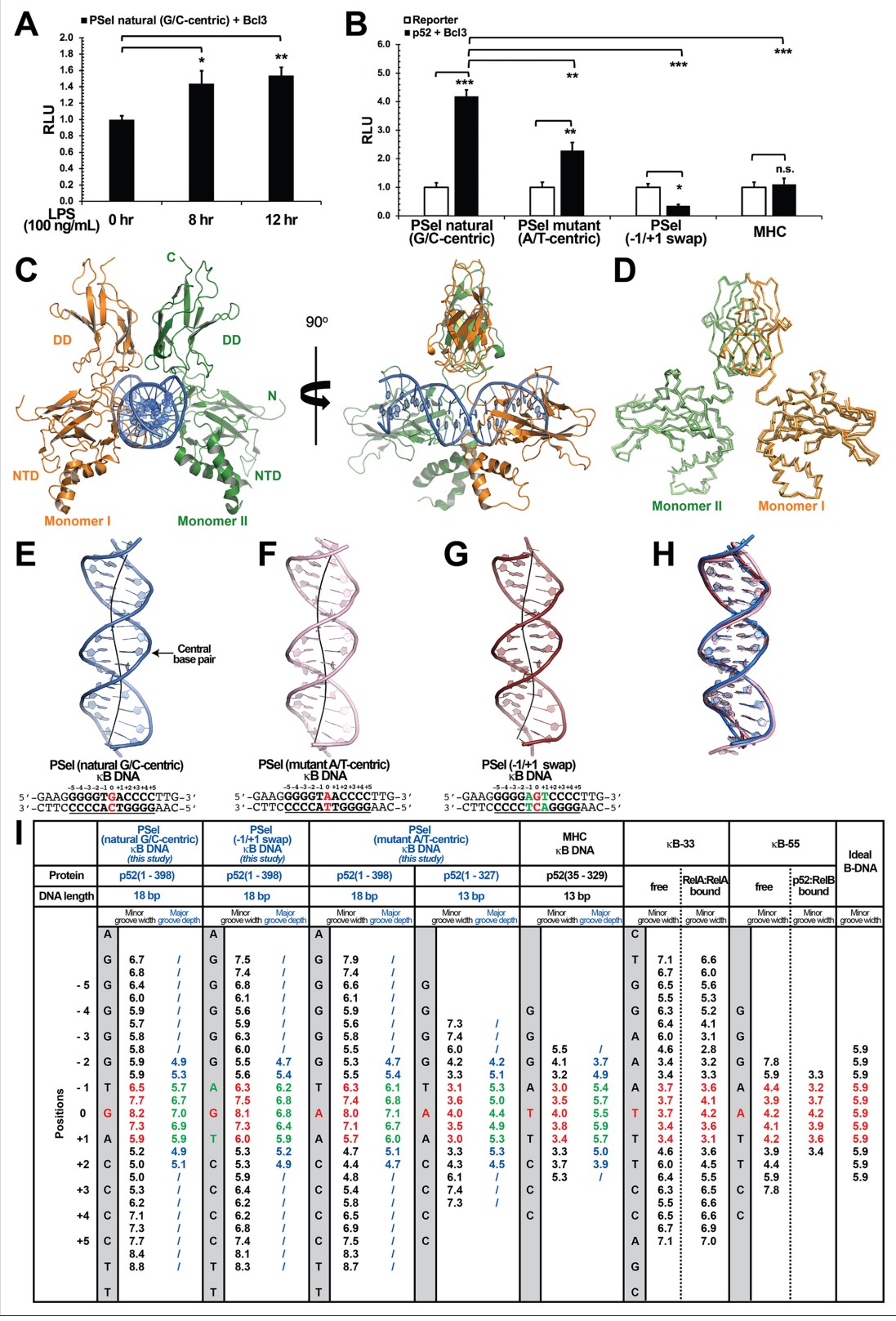

**Figure 1.** Crystal structures of p52:p52 homodimer in complex with PSel-κB DNA variants reveal distinct signatures. (**A**) The natural G/C-centric PSel luciferase reporter was activated by endogenous NF-κB with LPS stimulation and Bcl3 co-expression. The data were analyzed from three independent experiments performed in triplicate. RLU, relative luciferase unit. *p<0.05; **p<0.01 (t test). Error bars represent standard deviation (SD). (**B**) Luciferase reporter activity driven by co-expression of p52 and Bcl3 was reduced when the natural G/C-centric PSel site was mutated to A/T-centric or −1/+1 swap

*Figure 1 continued on next page*

*Figure 1 continued*

sites; and the MHC luciferase reporter was not activated by p52:p52:Bcl3 complex. The data were analyzed from three independent experiments performed in triplicate. *p<0.05; **p<0.01; ***p<0.001; n.s., not significant (t test). Error bars represent SD. (**C**) Overall structure of p52:p52 in complex with the natural G/C-centric PSel- κ B DNA. (Left) Ribbon diagram showing the entire complex viewed down the DNA helical axis. The two p52 monomers are shown in orange (monomer I) and green (monomer II), respectively; and the DNA duplex is shown in blue; (Right) View of the complex after rotating 90° along the vertical axis. (**D**) Overlay p52:p52 homodimers in three PSel- κ B DNA variants by their dimerization domain (DD). Monomer I is shown in tv_orange, bright orange, and light orange; monomer II is shown in forest, tv_green, and lime in the natural G/C-centric, mutant A/T-centric, and −1/+1 swap complexes. All three structures are presented as backbone traces. (**E–G**) Structure of the 18 bp PSel- κ B DNAs with (**E**) natural G/C-centric (blue), (**F**) mutant A/T-centric (light pink), and (**G**) −1/+1 swap (ruby). The DNA bps as observed in the co-crystal structures are shown in filled sticks. The view is onto the central minor groove. The nucleotide sequences used in co-crystallization are shown at the bottom, with  κ B DNA underlined and numbering scheme indicated above; the central position 0 is highlighted in red, and the swap of −1 and +1 positions is highlighted in green. (**H**) Overlay of natural G/C-centric, mutant A/T-centric, and−1/+1 swap PSel- κ B DNAs in (**E–G**). (**I**) Table showing minor groove widths and major groove depths (Å); the ideal B-form DNA was built using Coot program (*Emsley and Cowtan, 2004*; *Emsley et al., 2010*) based on the sequence of PSel- κ B DNA. The minor groove widths at the central region from position −1 to +1 are shown in red, and the corresponding major groove depths are shown in green. Geometrical parameters and the helical axes were calculated with Curves+. Groove widths are measured as minimal distances between the backbone spline curves passing through the phosphorus atoms; values are given at base pair levels and halfway between these levels (*Blanchet et al., 2011*; *Lavery et al., 2009*). LPS, lipopolysaccharide.

The online version of this article includes the following source data and figure supplement(s) for figure 1:

**Figure supplement 1.** p52 and DNA crystallization.

**Figure supplement 1—source data 1.** Raw image of SDS-PAGE gels in *Figure 1—figure supplement 1H*, with label.

**Figure supplement 1—source data 2.** Raw image of SDS-PAGE gels in *Figure 1—figure supplement 1H*, without label.

**Figure supplement 1—source data 3.** Raw image of SDS-PAGE gels in *Figure 1—figure supplement 1H*, with label.

**Figure supplement 1—source data 4.** Raw image of SDS-PAGE gels in *Figure 1—figure supplement 1H*, without label.

**Figure supplement 2.**  κ B DNA conformations.

the DDs reveals large rigid body movement of NTDs with rotation of ~20° and translation along rotation axis of ~1.4 Å (*Figure 2A*). This results in shifting of the NTD along PSel DNA toward its flanks by ~13 Å for both sides. In addition, the minor groove of the MHC-κB DNA at the central segment is compressed like all other NF-κB-DNA complexes indicated above (*Figure 1I*; *Figure 1—figure supplement 2C*).

The PSel-κB DNAs (18 bp) and recombinant p52 protein (aa 1–398, including the GRR) used in the current study are both longer than those in the MHC-κB DNA complex (13 bp and aa 35–329). In fact, all currently available structures of NF-κB-κB DNA complexes in the Protein Data Bank (PDB) contain short NF-κB proteins (only NTD and DD) and A/T-centric κB DNAs (*Figure 1—figure supplement 2E*; *Ghosh et al., 1995*; *Müller et al., 1995*; *Cramer et al., 1997*; *Chen et al., 1998a*; *Huang et al., 2001*; *Moorthy et al., 2007*; *Fusco et al., 2009*; *Chen et al., 1998b*). To test if the DNA and protein length variations induce structural changes in the complex, we set out to co-crystallizations of a shorter p52 protein (aa 1–327) with all three PSel-κB DNAs at various lengths (*Supplementary file 1*). This short p52 protein only co-crystallized with a 13 bp PSel (mutant A/T-centric) DNA but not the (natural G/C--centric) or (−1/+1 swap) DNAs in any lengths. The conformation of this complex is nearly identical to p52-MHC-κB complex with minor groove width less than 4 Å at the central position (*Figure 2B*; *Figure 1I*; *Figure 1—figure supplement 2D*). This crystal with short p52 is in a different crystal form compared to the three structures with the long p52, and it is also in a different crystal form compared to the MHC-κB DNA complex, suggesting that crystal packing is unlikely to be the main cause of the structural differences, and that both the DNA and protein lengths play significant roles.

Therefore, we observed an influence of the length of the p52 protein on the conformation of the κB DNA and the organization of the p52:p52 dimer in the complex. The length and sequence of DNA also influence the structures, and we believe they are correlated with the length of p52 protein. As discussed above, the short p52 protein (aa 1–327) failed to interact with Bcl3 (*Figure 1—figure supplement 1E–H*), partly due to the lack of the GRR. We used the long p52 protein (aa 1–398) for the rest of the studies.

## Distinct protein-DNA interactions in the (p52:p52)-DNA complexes

The widening of the minor groove propagates from the central position to all four base steps on both sides with values around 5–6 Å (*Figure 1I*). This widening and the consequent deepening of the major

**Table 1.** Summary of crystallographic information.

| Structure | p52-PSel (natural G/C-centric) (7CLI) | p52-PSel (mutant A/T-centric) (7VUQ) | p52-PSel (−1/+1 swap) (7VUP) | p52-PSel (mutant 13-mer A/T-centric) (7W7L) |
|---|---|---|---|---|
| p52 construct | aa 1–398 | aa 1–398 | aa 1–398 | aa 1–327 |
| DNA length | 18 bp | 18 bp | 18 bp | 13 bp |
| **Data collection** | | | | |
| Wavelength (Å) | 0.97852 | 0.979183 | 0.979183 | 0.979183 |
| Resolution range (Å) | 45.63–3.00 (3.16–3.00) | 46.48–3.10 (3.27–3.10) | 46.85–3.40 (3.58–3.40) | 48.77–3.00 (3.16–3.00) |
| Space group | $P2_12_12_1$ | $P2_12_12_1$ | $P2_12_12_1$ | $P6_222$ |
| $a, b, c$ (Å) | 84.50, 85.37, 140.29 | 84.45, 84.63, 139.43 | 83.99, 84.29, 140.57 | 225.28, 225.28, 96.94 |
| $\alpha, \beta, \gamma$ (°) | 90, 90, 90 | 90, 90, 90 | 90, 90, 90 | 90, 90, 90 |
| Mosaicity (°) | 0.54 | 0.2 | 0.38 | 0.17 |
| Total no. of reflections | 279,204 (42,196) | 246,551 (36,406) | 185,745 (27,383) | 1,101,703 (164,545) |
| No. of unique reflections | 19,950 (2963) | 18,756 (2690) | 14,272 (2025) | 29,500 (4217) |
| Completeness (%) | 95.4 (98.2) | 99.8 (99.6) | 99.7 (99.9) | 100 (100) |
| Multiplicity | 14.0 (14.2) | 13.1 (13.5) | 13.0 (13.5) | 37.3 (39.0) |
| Mean $I/\sigma(I)$ | 14.3 (1.9) | 16.1 (2.1) | 9.6 (1.6) | 18.0 (2.4) |
| $R_{p.i.m.}$ | 0.037 (0.498) | 0.030 (0.455) | 0.049 (0.618) | 0.031 (0.396) |
| CC1/2 | 0.999 (0.767) | 0.999 (0.934) | 0.998 (0.782) | 0.999 (0.840) |
| Wilson B-factor (Å²) | 86.8 | 100.1 | 118 | 98.9 |
| **Refinement** | | | | |
| Resolution range (Å) | 42.72–3.00 (3.29–3.00) | 40.00–3.10 (3.39–3.10) | 45.45–3.40 (3.72–3.40) | 48.77–3.00 (3.29–3.00) |
| No. of reflections | 19,898 | 17,720 | 13,543 | 27,943 |
| $R_{work}/R_{free}$ | 0.236/0.275 | 0.224/0.249 | 0.271/0.286 | 0.239/0.258 |
| No. of non-hydrogen atoms | 5359 | 5381 | 5359 | 5191 |

*Table 1 continued on next page*

*Table 1 continued*

| Structure | p52-PSeI (natural G/C-centric) (7CLI) | p52-PSeI (mutant A/T-centric) (7VUQ) | p52-PSeI (−1/+1 swap) (7VUP) | p52-PSeI (mutant 13-mer A/T-centric) (7W7L) |
|---|---|---|---|---|
| Protein | 4634 | 4668 | 4668 | 4640 |
| DNA | 713 | 713 | 691 | 527 |
| Water | 12 | | | 24 |
| Average B factors (Å²) | | | | |
| Protein | 115.3 | 139.6 | 165.6 | 128.9 |
| DNA | 116.7 | 141.2 | 178.5 | 80.1 |
| Water | 63.2 | | | 81.2 |
| R.m.s. deviations | | | | |
| Bond length (Å) | 0.012 | 0.011 | 0.010 | 0.011 |
| Bond angles (°) | 1.78 | 1.866 | 1.735 | 1.887 |
| Ramachandran plot (%) | | | | |
| Favored | 96 | 95.24 | 93.54 | 93.15 |
| Allowed | 4 | 4.76 | 5.78 | 6.85 |
| Outliers | | | 0.68 | |

The numbers in parentheses are for the highest resolution shell. One crystal was used for each data collection.

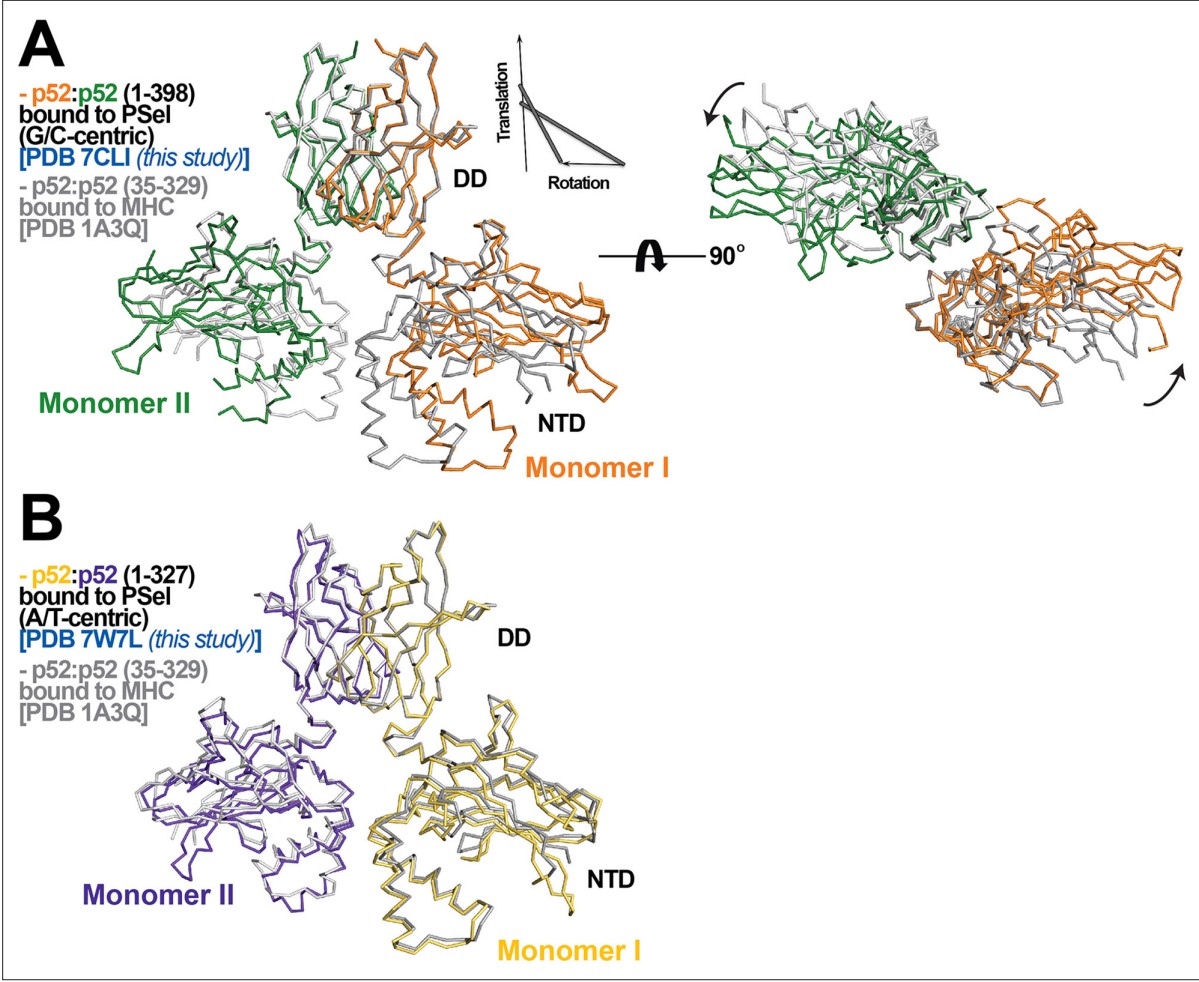

**Figure 2.** p52:p52 dimer conformations. (**A**) (Left) Overlay of p52:p52 (aa 1–398) in complex with the 18 bp natural G/C-centric PSel-*κ*B DNA (PDB 7CLI, this study) (in orange and green for monomers I and II, respectively) and p52:p52 (aa 35–329) in complex with the MHC-*κ*B DNA (PDB 1A3Q) (in gray). Diagram explains rigid-body movement of the NTD. (Right) View of the complex after rotating 90° along the horizontal axis. Both protein structures are presented as backbone traces. (**B**) Overlay of the short p52:p52 (aa 1–327) in complex with the 13 bp PSel (mutant A/T-centric)-*κ*B DNA (PDB 7W7L, this study) (in yellow orange and purple blue for monomers I and II, respectively) and p52:p52 (aa 35–329) in complex with the MHC-*κ*B DNA (PDB 1A3Q) (in gray).

groove have significant impact on protein-DNA interactions. The most significant of which is the loss of cross-strand base contacts by Arg52 (*Figure 3A*). The cross-strand contacts between the homologous Arg (Arg54 in p50 and Arg33 in RelA) and DNA are observed in all other A/T-centric NF-κB-DNA structures (*Figure 3B*; *Supplementary file 3*). In the 18 bp long PSel-κB DNA complexes, the NH1 and NH2 groups of Arg52 form H-bonds with both O6 and N7 groups of G at ±3. The same Arg52 in the complex with the short 13 bp PSel (mutant A/T-centric) or the MHC-κB DNA only hydrogen bonds with the O6 group, but not the N7, of G at ±3. The homologous Arg54 in p50 also contacts the O6 group of G at −3 in the p50:RelA-IFNβ-κB DNA complex.

The other notable feature of the PSel-κB DNA complexes is the highly asymmetric DNA contacts by p52:p52 homodimer. Monomer I is closer to its cognate half-site making more direct contacts with the DNA than monomer II (*Figure 3—figure supplement 1A*). Although asymmetric DNA binding by the symmetric homodimer is a common feature in all NF-κB-DNA complexes, it is significantly more pronounced in the present structures. Moreover, the p52:52 homodimer also displays substantial asymmetry. With the DD of the two monomers in superposition, the NTDs rotate from each other by ~6° (*Figure 3C*). The interdomain interaction is extensive in monomer I compared to that in monomer II (*Figure 3D*).

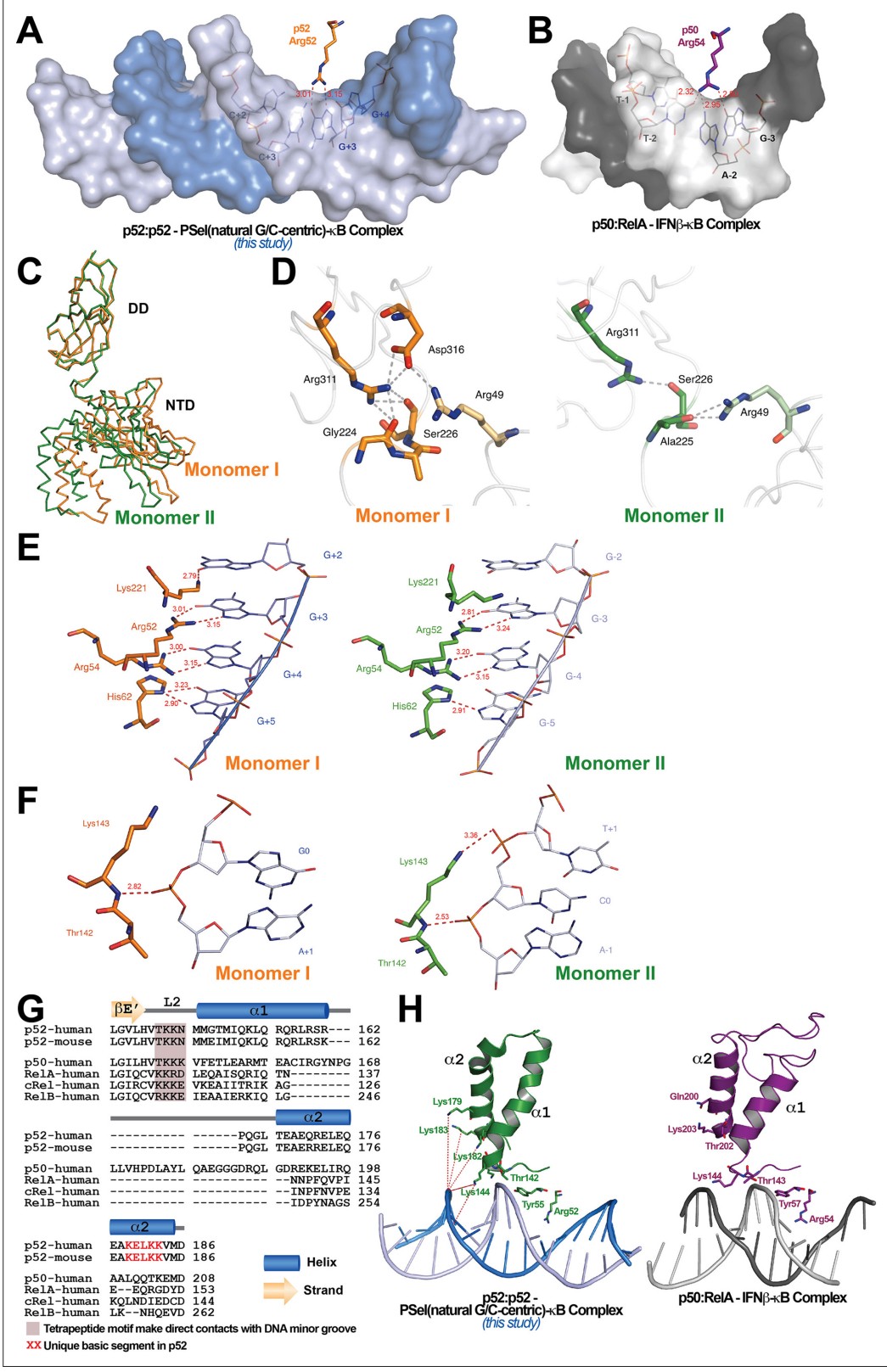

**Figure 3.** Protein-DNA contacts. (**A**) Arg52 of p52 in the PSel- $\kappa$ B complex (PDB 7CLI, this study) only makes base-specific contacts with G at +3 position. (**B**) The corresponding Arg54 of p50 in the p50:RelA-IFNb- $\kappa$ B complex (PDB 1LE5) makes base-specific contacts with A at −2 and G at −3 positions as well as additional cross-strand contacts with T at −2 position. (**C**) Conformational differences between two p52 monomers in complex

*Figure 3 continued on next page*

*Figure 3 continued*

with the 18 bp natural G/C-centric PSel-κB DNA (PDB 7CLI, this study) are shown by superposing their DDs. The two monomers are presented as backbone traces. (**D**) Hydrogen bonding network at the interdomain interface between DD and NTD in each p52 monomer. In monomer I, the two domains form contacts with each other through a wide network of H-bonds between the side chains of Arg49 from the NTD and Gly224, Ser226, Arg311, and Asp316 from the DD; whereas in monomer II, there are only contacts between Arg49 and Ala225, as well as Ser226 and Arg311. Residues from NTD are colored in lighter orange and lighter green for monomers I and II, respectively. (**E**) DNA based-specific contacts made by Arg52, Arg54, His62, and Lys221 of p52 (Left) monomer I and (Right) monomer II in complex with the natural PSel-κB DNA. H-bonds are indicated as red dotted lines with distances labeled. Noted that Lys221 in monomer II is in a different conformation and has no specific contacts with DNA. (**F**) DNA backbone contacts made by Lys143 of p52 (Left) monomer I and (Right) monomer II. (**G**) Sequence alignment showing the unique basic segment in p52 among all NF-κB family members. Both human and mouse sequences of the p52 subunit are shown. Only human sequences are shown for the rest of the family members. Secondary structures and connecting loops are drawn above the sequences. (**H**) (Left) The unique basic segment in p52 NTD helix α2 interacts with PSel-κB DNA in the present structure (PDB 7CLI, this study); (Right) these interactions are absent in p50 subunit in the p50:RelA-IFNb-κB complex (PDB 1LE5).

The online version of this article includes the following figure supplement(s) for figure 3:

**Figure supplement 1.** Asymmetric p52 monomers.

In the PSel-κB complex, the side chains of Lys221, Arg52, Arg54, and His62 in p52 monomer I make direct base-specific contacts to four consecutive G(s) from +2 to +5 positions (*Figure 3E*, left). In addition, Ser61 also makes direct contact with A at ±6 and ±7 positions; these contacts are not possible for the short p52 (aa 1–327) co-crystallized with 13 bp κB DNAs such as MHC and PSel (mutant A/T-centric)-κB (*Figure 3—figure supplement 1A–B*). p52 monomer II makes contact with only three G(s) from position −3 to −5 (*Figure 3E*, right). The conformation of loop L3 in the two p52 monomers is different; consequently, only Lys221 in monomer I makes specific contacts with G at position +2 (*Figure 3—figure supplement 1C*). Glu58 helps to position Arg52, Arg54, and His62, and makes base-specific interaction to the opposite C at ±3 positions.

In addition to base-specific interactions, there are multiple protein contacts to the DNA phosphate backbone, mostly to the central region of the DNA. The side chain of Cys57 hydrogen bonds to the backbone phosphate group of C at ±2; and the side chains of Tyr55 and Lys143 hydrogen bond to the backbone phosphate group of A at ±1 (*Figure 3—figure supplement 1A*). Only in monomer II does the side chains of Lys143 make an additional H-bond to the backbone phosphate group of C at position 0 (*Figure 3F*). Interestingly, all other NF-κB-DNA complexes, including the short p52:p52 homodimer bound to both 13 bp MHC-κB and PSel (mutant A/T-centric)-κB DNAs, exhibit more backbone contacts by Gln284 and Gln254 (*Figure 3—figure supplement 1B*; *Supplementary file 3*). The presence of an additional positively charged residue in loop L2 in the other NF-κB subunits (p52: T$^{142}$KKN; p50: TKKK; and RelA: KKRD) enhances backbone binding at the minor groove side including cross-strand interactions (*Figure 3G*). In addition, there is a unique basic segment in p52, a peptide-rich in basic residues (K$^{179}$ELKK), located near the end of helix α2 (*Figure 3G*). These basic residues possibly mediate long-range electrostatic interactions with the negatively charged DNA backbone which might pull the DNA strands away from each other toward the p52 protein (*Figure 3H*). In summary, aa composition in loop L2 and helix α2, might play an important role in determining DNA binding by the NF-κB dimers.

## MD simulations reveal free DNAs exhibit distinct preferred conformations

In order to investigate whether the minor groove width of the PSel-κB DNA variants observed in the current complexes is induced by the protein or intrinsic to DNA sequences, we first carried out microsecond MD simulations of the four κB DNAs in free form. The simulations were initiated using DNA conformations in the crystal structures of the complexes, where the three PSel-κB DNA variants had a widened minor groove and the MHC-κB DNA a narrow minor groove. Throughout our simulations PSel (natural G/C-centric) largely maintained its widened minor groove at the central 0 position, while the minor groove in PSel (mutant A/T-centric) narrowed slightly; PSel (−1/+1 swap) displayed a significantly narrowed minor groove, which became similar to that of MHC-κB DNA recorded in the

simulations (*Figure 4A*). The swap of T and A at ±1 positions reverses the geometric conformation of bps at both positions (*Figure 4C*). Specifically, these swaps exchange pyrimidines and purines at ±1 positions, forcing the swapped bps to adopt an opposite shear and buckle direction to optimize base stacking with neighboring bps compared to those on the non-swapped DNAs. The thymines at both positions slide and tilt toward the minor groove simultaneously, narrowing the central minor grooves (*Figure 4—figure supplement 1A*).

The simulations also reveal a narrowed minor groove of the A/T-centric DNA at +1 position compared to the corresponding G/C-centric DNA with the same flanking bps, that is, PSel (mutant A/T-centric) compared to PSel (natural G/C-centric), and MHC compared to PSel (−1/+1 swap) (*Figure 4B*). The A:T bp at 0 position shows large shear, buckle, and opening, forcing a register toward the minor groove in the curvature of free A/T-centric DNAs (*Figure 4—figure supplement 1A*). With only two H-bonds, A/T steps can easily unwind to yield low propeller twist. The neighboring A:T at 0 and +1 position in PSel (mutant A/T-centric) (0 and −1 position in MHC) rotate the adenines likewise to optimize base stacking (*Figure 4C*). This conformation bends the minor groove and can further deform the local structure of PSel (mutant A/T-centric) through the occasionally formed cross-strand H-bonds between the neighboring steps (*Figure 4—figure supplement 2*). It appears that in MHC, the same narrowing effect introduced by ±1 swap fixed the twisted conformation at 0 position, thereby stabilizing its local structure instead. Collectively, we conclude that having successive A:T is likely to reduce the intrinsic minor groove width. Our finding is in line with the observations of a compressed minor groove of free κB-33 DNA or the bending into the minor groove from continuous A:T in A-tract DNAs (*Barbic et al., 2003*; *Huang et al., 2005*).

Comparison of MD simulations and crystal structures suggests that upon binding to p52, the −1/+1 swap DNA experiences more disruptive conformational change than the natural G/C-centric PSel-κB (*Figure 4A*; *Figure 4—figure supplement 1B*). The binding at the central part of both DNAs is symmetrically facilitated through the H-bonds between two residues (Lys143 and Tyr55) from each monomer of p52 and the phosphate of nucleotides at −1 and +1 positions as well as the T-shaped π-stacking interactions between DNA bases and Tyr55. The aforementioned two residues in the two p52 monomers are positioned further apart in the PSel-κB DNA complex compared to the MHC-κB DNA complex. Unlike the natural G/C-centric DNA, the binding on both strands of −1/+1 swap DNA draws the bound thymine in the opposite directions of the minor groove, breaking the intra-bp H-bonds and severely distorting the central bps (*Figure 4—figure supplement 1B*). It appears that p52:p52 homodimer adopts a specific conformation with a nearly fixed inter-monomer distance upon the binding to κB DNAs, such that it tears the central part of −1/+1 swap DNA into a favored binding conformation. Overall, the comparative analysis of MD simulations and crystal structures suggests that the p52:p52 homodimer induces the least amount of conformational changes on κB DNA with an intrinsically widened minor groove.

## The p52 homodimer recognizes κB DNAs with different thermodynamic features

Structural analysis described above did not provide a strong correlation between the conformational states of (p52:p52)-DNA complexes and the transcriptional output. We next tested whether p52:p52 homodimer binds to the natural G/C-centric, mutant A/T-centric, and −1/+1 swap PSel-κB, as well as MHC-κB DNAs with different mechanisms and/or affinities. In all cases, long p52 protein (aa 1–398) as well as long DNA were used. Isothermal titration calorimetry (ITC) reveals that p52:p52 binds all three PSel-κB DNA variants with similar binding affinities ($K_d$) in the range of approximately 40–90 nM whereas it binds MHC-κB DNA much tighter (*Figure 5*; *Figure 5—figure supplement 1*). However, binding of p52 to the natural G/C-centric PSel-κB DNA is associated with a large increase in entropy (ΔS) and a moderate decrease in enthalpy (ΔH). On the other hand, the binding to the MHC and mutant A/T-centric PSel DNAs showed a much larger decrease in enthalpy. These results suggest that the binding of p52:p52 homodimer to the G/C-centric κB DNA is favored more by entropy, whereas the binding to the A/T-centric DNA is driven by enthalpy alone.

To test if this mechanism is general to other κB DNAs, we also determined the thermodynamic parameters for p52 binding to Skp2-κB DNA. Skp2-κB DNA is present in the promoter of S-phase kinase-associated protein 2 (Skp2) and it is also regulated by the p52:p52 homodimer and Bcl3 (*Figure 5—figure supplement 2A*; *Barré and Perkins, 2007*; *Wang et al., 2012*). Skp2-κB DNA is

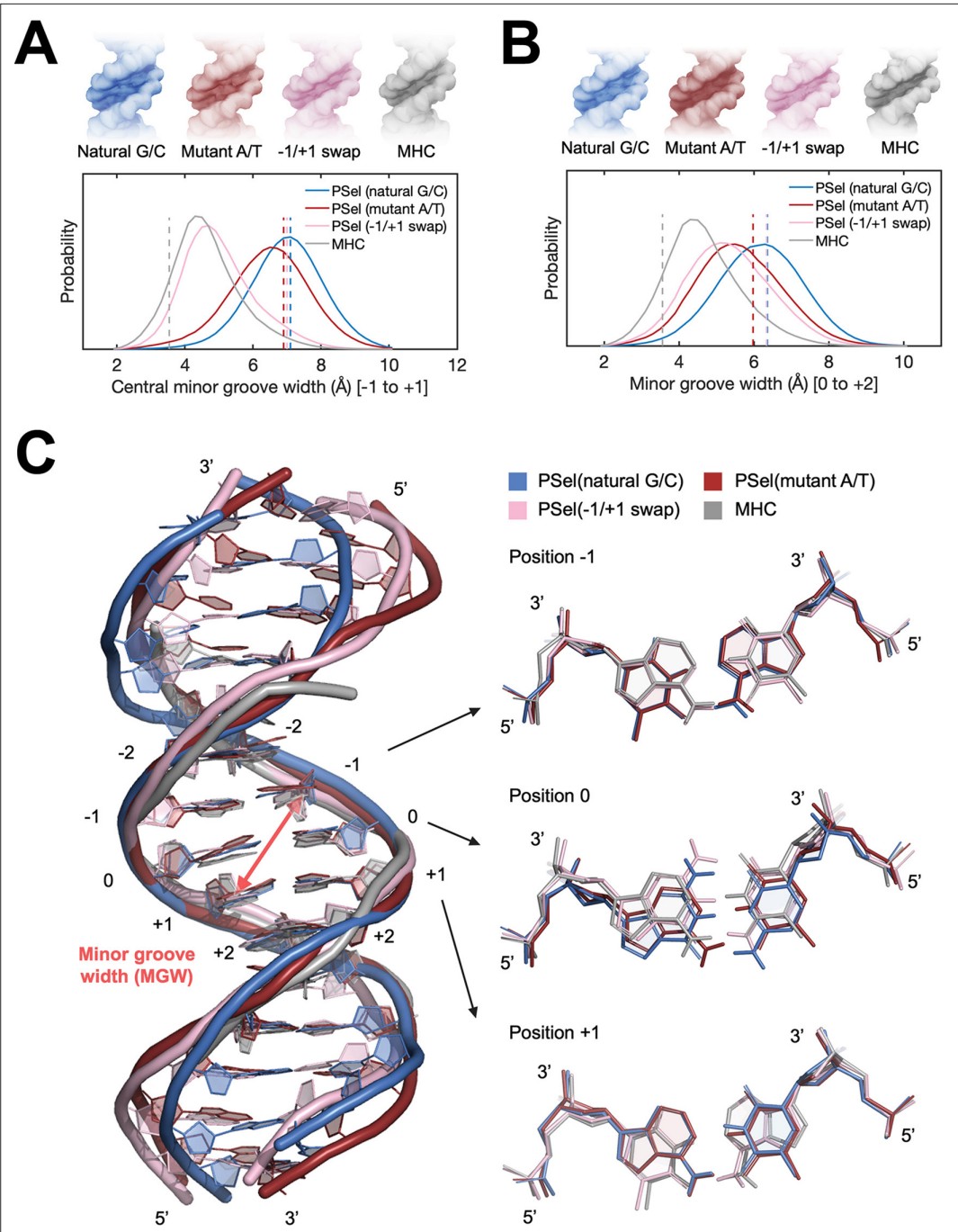

**Figure 4.** MD simulations for free $\kappa$ B DNAs. (**A, B**) Statistics of the minor groove width over the aggregated 10-μs simulations of each system at (**A**) the central 0 position (averaged over the five levels from −1 to +1 positions) and (**B**) the +1 position (averaged over the five levels from 0 to +2 positions). (Upper) DNA isosurface at 0.2 isovalue (20% occupancy); (Lower) Probability distribution of minor groove width. Dashed lines show the minor groove width in the (p52:p52)-bound crystal structures. (**C**) Representative structures of natural G/C-centric, mutant A/T-centric, −1/+1 swap PSel-$\kappa$ B DNAs, and MHC-$\kappa$ B DNA revealed from MD simulations. (Left) Superimposed structures showing the narrowed central minor groove on −1/+1 swap DNAs; (Right) Representative conformations of bps at −1, 0, and +1 positions revealed from MD simulations. Minor groove width was calculated with Curves+ (***Blanchet et al., 2011***; ***Lavery et al., 2009***). MD, molecular dynamics.

The online version of this article includes the following figure supplement(s) for figure 4:

**Figure supplement 1.** Free DNA simulations.

**Figure supplement 2.** DNA conformations in MD simulations.

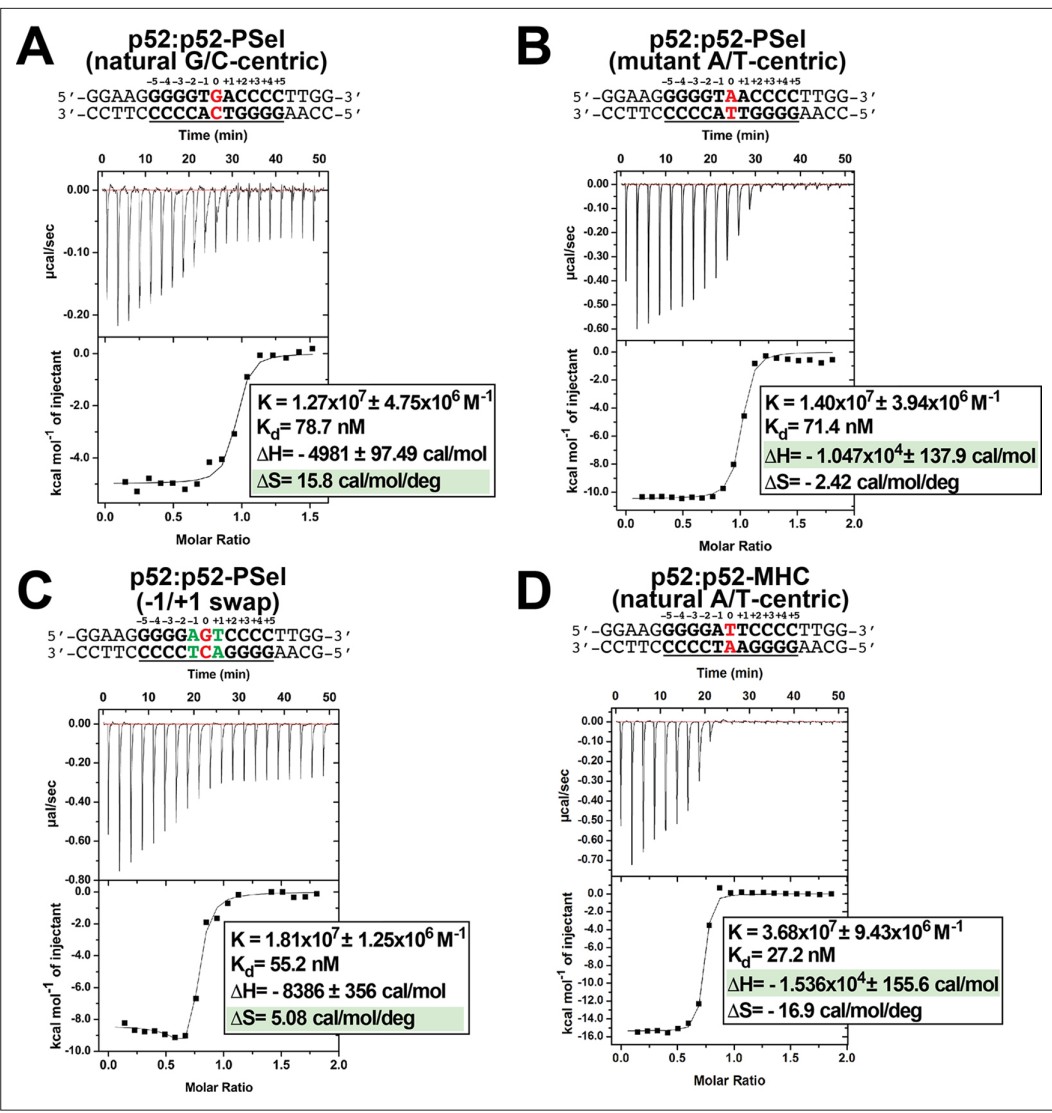

**Figure 5.** p52 binds $\kappa$B DNAs with different thermodynamic features. (**A–D**) Calorimetric titration data showing the binding of recombinant p52:p52 (aa 1–398) homodimer with (**A**) natural G/C-centric, (**B**) mutant A/T-centric, (**C**) −1/+1 swap PSel-$\kappa$B, and (**D**) MHC-$\kappa$B DNAs. The top panel of the ITC figures is the raw data plot of heat flow over time for the titration of 300 μM of indicated DNA into 35 μM of p52; the bottom panel shows the corresponding plot after integration of peak areas and normalization to yield a plot of ΔH against the DNA/p52 ratio and the line represents the best fit to the data according to a single-site binding model. The determined $K_d$, changes of enthalpy and entropy are shown on the bottom panel. ITC, isothermal titration calorimetry.

The online version of this article includes the following figure supplement(s) for figure 5:

**Figure supplement 1.** Repeat ITC experiments in *Figure 5*.

**Figure supplement 2.** p52 interacts with Skp2-$\kappa$B DNA.

another natural G/C-centric (5′-GGGG<u>AGT</u>TCC-3′) κB DNA but with the presence of A:T and T:A bp at +1 and −1 positions, the same as the −1/+1 swap PSel DNA in the central region. Skp2 also has a very different 4 bp half-site, TTCC, at the +1 to +4 positions. The $K_d$ as well as relative contributions of entropy and enthalpy to the binding to Skp2 and −1/+1 swap PSel DNAs are similar (*Figure 5C*; *Figure 5—figure supplement 1C*, *Figure 5—figure supplement 2B*). These results suggest that DNA sequence and conformational differences lead p52 to bind DNAs through different thermodynamic binding processes. However, thermodynamic binding mechanism does not fully capture the differential transcriptional output mediated by these κB DNAs.

## The p52 homodimer binds κB DNAs with different kinetic features

We next examined the binding kinetics of p52 and κB DNAs as there is mounting evidence that, separate from binding affinity, kinetic rate constants ($k_{on}$ and $k_{off}$) are crucial to the physiological effects of protein-ligand interactions in a variety of cellular processes (*Nakajima et al., 2001*; *Gross and Lodish, 2006*; *González et al., 2005*; *Markgren et al., 2002*). We utilized biolayer interferometry (BLI) to study the association and dissociation rate of p52:p52 binding to various κB DNAs. Biotinylated DNAs were immobilized on the streptavidin (SA) sensors and tested with purified p52 protein. The binding kinetics differ significantly among the DNAs. The binding of more transcriptionally active natural G/C-centric PSel showed a higher association ($k_{on}$) and dissociation rate ($k_{off}$) than the other two variants and MHC-κB DNAs (*Figure 6A–E*). Consistently, in the case of Skp2-κB DNA, the more transcriptionally active natural G/C-centric Skp2 showed a faster kinetics than its mutant A/T-centric, especially the $k_{off}$ (*Figure 6—figure supplement 1A–C*).

We further determined the $k_{on}$ and $k_{off}$ of the transcriptionally competent p52:p52:Bcl3 complex binding to PSel-κB DNA variants by BLI. In agreement with our previous study, only the recombinant phospho-mimetic Bcl3 from *Escherichia coli* forms ternary complex with p52:p52 homodimer and κB DNA (*Wang et al., 2017*). Both recombinant WT and phospho-mimetic Bcl3 protein interact with p52 with similar kinetics (*Figure 1—figure supplement 1F*, *Figure 6—figure supplement 2A–C*); however, the p52:p52:WT-Bcl3 complex does not bind DNAs (*Figure 6—figure supplement 2D–E*). The binding of p52:p52:phospho-Bcl3 with the natural G/C-centric PSel DNA exhibited both higher $k_{on}$ and $k_{off}$ (*Figure 6F–J*). Similarly, the binding with the natural G/C-centric Skp2 DNA also showed a higher $k_{off}$ comparedto its A/T-centric mutant (*Figure 6—figure supplement 1D–F*).

Overall, the binding kinetics of p52:p52 homodimer alone versus p52:p52:Bcl3 complex follows the same trend. Moreover, a comparison of binding affinity, association, and dissociation rates with respect to the more transcriptionally active PSel and Skp2-κB sites shows a correlation between transcriptional output and the dissociation rate. The slower the $k_{off}$, the lower the reporter activities for both (p52:p52)-DNA and (p52:p52:Bcl3)-DNA complexes (*Figure 6K*; *Figure 6—figure supplement 1G*). Therefore, transcriptional activity may have a closer link to the binding kinetics rather than the thermodynamic stability of the complex.

## Differential minor groove geometries correlate with differential DNA binding kinetics

To better understand how binding kinetics correlates with DNA sequences, we carried out four 3-μs MD simulations for each of the three (p52:p52)-PSel-κB DNA complexes, including the natural G/C-centric, mutant A/T-centric, and −1/+1 swap DNAs. The overall conformations of p52:p52 dimers and DNAs are similar during the MD simulations, both of which are slightly more stable in the p52-natural G/C-centric DNA complex (*Figure 7—figure supplement 1A–C*). Upon p52:p52 homodimer binding, the central minor grooves of all three DNAs narrowed, resulting in a consistent trend as seen in the free form (*Figure 7—figure supplement 1D*). Adapting to the width of the DNA central minor groove, the p52:p52 subunits bound to −1/+1 swap DNA adopt a closed conformation where the two segments bound to the central minor groove appear to clamp the DNA, in contrast to their more open conformation when bound to the natural G/C-centric DNA; the p52 subunits bound to the mutant A/T-centric DNA fall in between the aforementioned two types of conformations (*Figure 7—figure supplement 2*). Notably, we found that the NTD of p52:p52 subunits could make contacts across the central minor grooves when bound to mutant A/T-centric and −1/+1 swap DNAs (*Figure 7A*). Specifically, the clamping conformation enables Lys144 to form cross-strand binding that engages the phosphates at −3/+3 positions in the opposite DNA strands in the mutant A/T and −1/+1 swap DNAs (*Figure 7B*). In the case of the −1/+1 swap DNA, the main chain amines of Lys144 in both monomers also retain contacts to the phosphates at −1/+1 positions in the nearby strand, likely due to the matching lysine side chain length and the narrowed minor groove width (*Figure 7B–C*). These cross-strand contacts, while recorded in all three protein-DNA complexes, were most frequently observed in the −1/+1 swap and mutant A/T-centric DNAs. Specifically, they were approximately five and three times more frequent than in the natural G/C-centric DNA, respectively (*Figure 7D*). This difference suggests that the dynamism of the p52:p52 homodimer varies as it recognizes specific minor groove geometries of the three PSel-κB DNAs. The DNA binding domains remain open preventing Lys144 from reaching out to the other DNA strand in the natural G/C-centric DNA. In the case of the

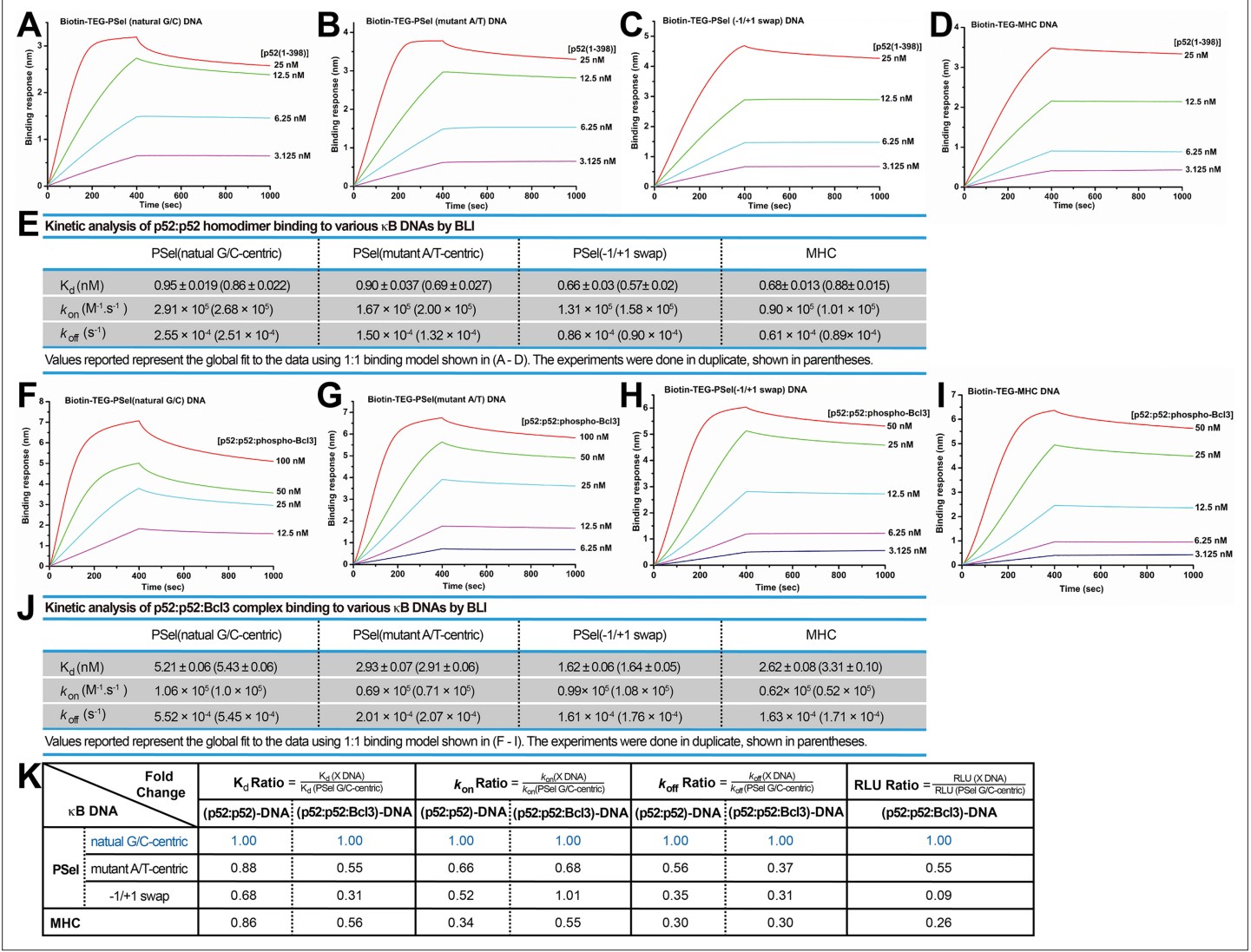

**Figure 6.** p52 binds the natural transcriptionally active G/C-centric PSel- $\kappa$ B DNA with faster kinetics. (**A–D**) Biolayer interferometry (BLI) binding analysis of p52:p52 (aa 1–398) homodimer to immobilized biotin labeled (**A**) natural G/C-centric, (**B**) mutant A/T-centric, (**C**) −1/+1 swap PSel- $\kappa$ B, and (**D**) MHC- $\kappa$ B DNAs. The differences in $k_{on}$ and $k_{off}$ can be seen in the shapes of the association and dissociation curves. Each experiment was done in duplicate and one representative set of curves is shown. (**E**) Table showing the kinetic analysis in (**A–D**). (**F–I**) BLI binding analysis of p52:p52:Bcl3 complex to immobilized biotin labeled (**F**) natural G/C-centric, (**G**) mutant A/T-centric, (**H**) −1/+1 swap PSel- $\kappa$ B, and (**I**) MHC- $\kappa$ B DNAs. Each experiment was done in duplicate and one representative set of curves is shown. (**J**) Table showing the kinetic analysis in (**F–I**). (**K**) Table summarizing the fold change of $K_d$, $k_{on}$, and $k_{off}$ with respect to the more transcriptionally active G/C-centric PSel- $\kappa$ B DNA. The average values of the duplicated kinetics data in (**A–J**) and the relative reporter activities in RLU from *Figure 1A–B* were used for ratio calculations. The numbers for the greater reporter active G/C-centric PSel are shown in blue.

The online version of this article includes the following source data and figure supplement(s) for figure 6:

**Figure supplement 1.** p52 and Skp2- $\kappa$ B DNA binding kinetics.

**Figure supplement 2.** Recombinant phospho-mimetic Bcl3 forms a ternary complex with p52 and  $\kappa$ B DNA.

**Figure supplement 2—source data 1.** Raw image of SDS-PAGE gel in *Figure 6—figure supplement 2A*, with label.

**Figure supplement 2—source data 2.** Raw image of SDS-PAGE gel in *Figure 6—figure supplement 2A*, without label.

−1/+1 swap DNA, the two p52 NTDs adopt a closed conformation allowing Lys144 to make cross-strand contacts which are further assisted by the DNA conformational change. On the one hand, such cross-strand binding provides extra contacts between the p52:p52 homodimer and the DNA, which may hinder their dissociation. On the other hand, the induced closing of NTDs may be unfavored by the p52:p52 homodimer and slow down the protein-DNA association.

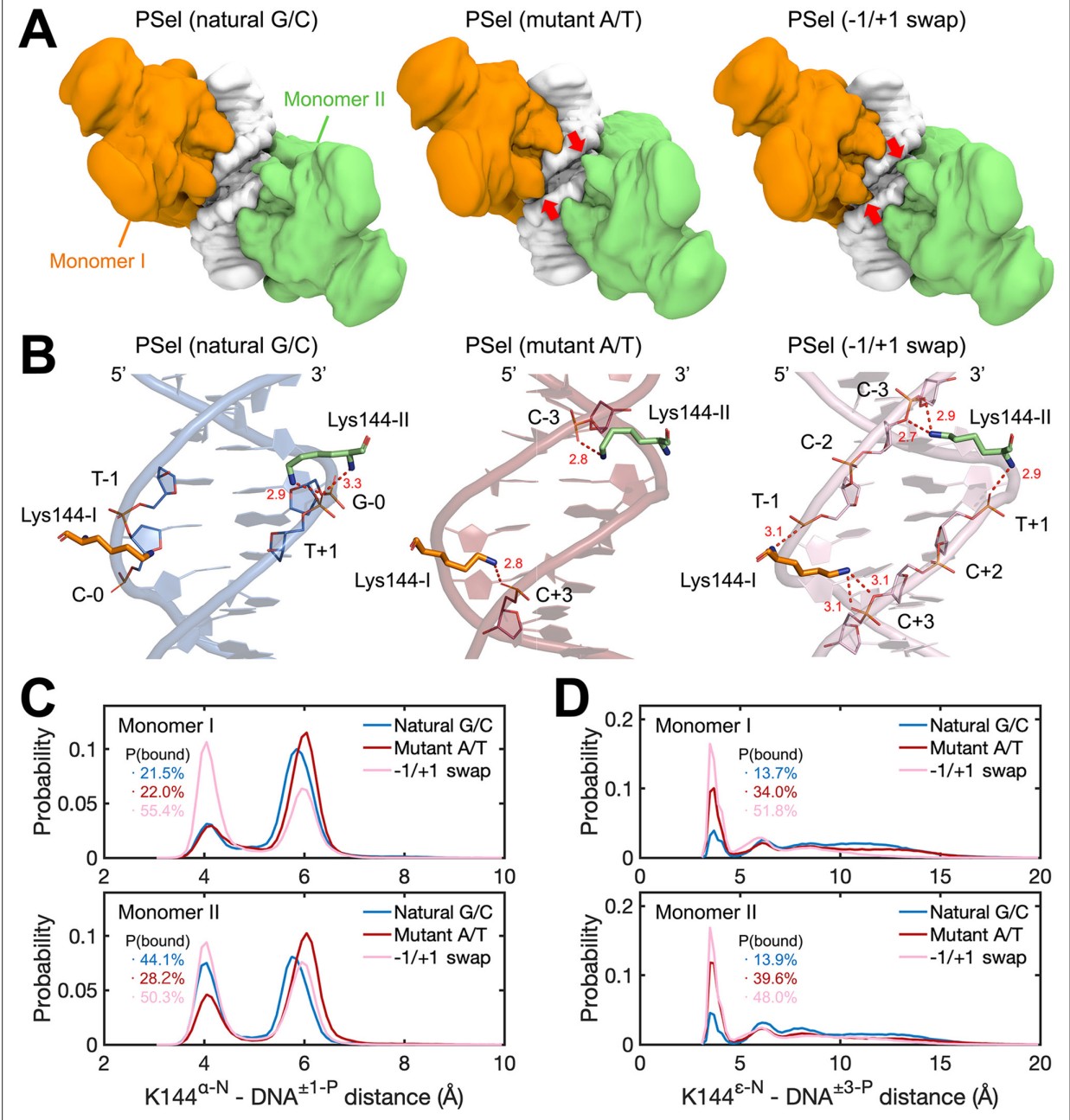

**Figure 7.** MD simulations for (p52:p52)-DNA complexes. (**A**) Isosurface of (p52:p52)-DNA complexes (20% occupancy). Red arrows indicate the cross-strand contacts by p52 homodimer observed in conformations bound to mutant A/T-centric and −1/+1 swap PSel- $\kappa$ B. (**B**) Representative binding conformations of Lys144 at the minor grooves showing the cross-strand binding formed in mutant A/T-centric and −1/+1 swap PSel- $\kappa$ B. Red dashed lines stand for the hydrogen bonds formed between Lys144 and DNAs. (**C**) Probability distributions of minimum distance between the α-N of Lys144 backbone and the P atom at position −1/+1 of the nearby strand of DNA, and (**D**) between the ε-N of Lys144 side chain and the P atom at position +3/−3 of the far strand of DNA. The cumulative probabilities of close contacts (<4.8 Å) are marked in the same coloring scheme. MD, molecular dynamics.

The online version of this article includes the following figure supplement(s) for figure 7:

**Figure supplement 1.** Conformation of (p52:p52)-DNA complexes in MD simulations.

**Figure supplement 2.** Binding conformation of p52:p52 homodimer in MD simulations.

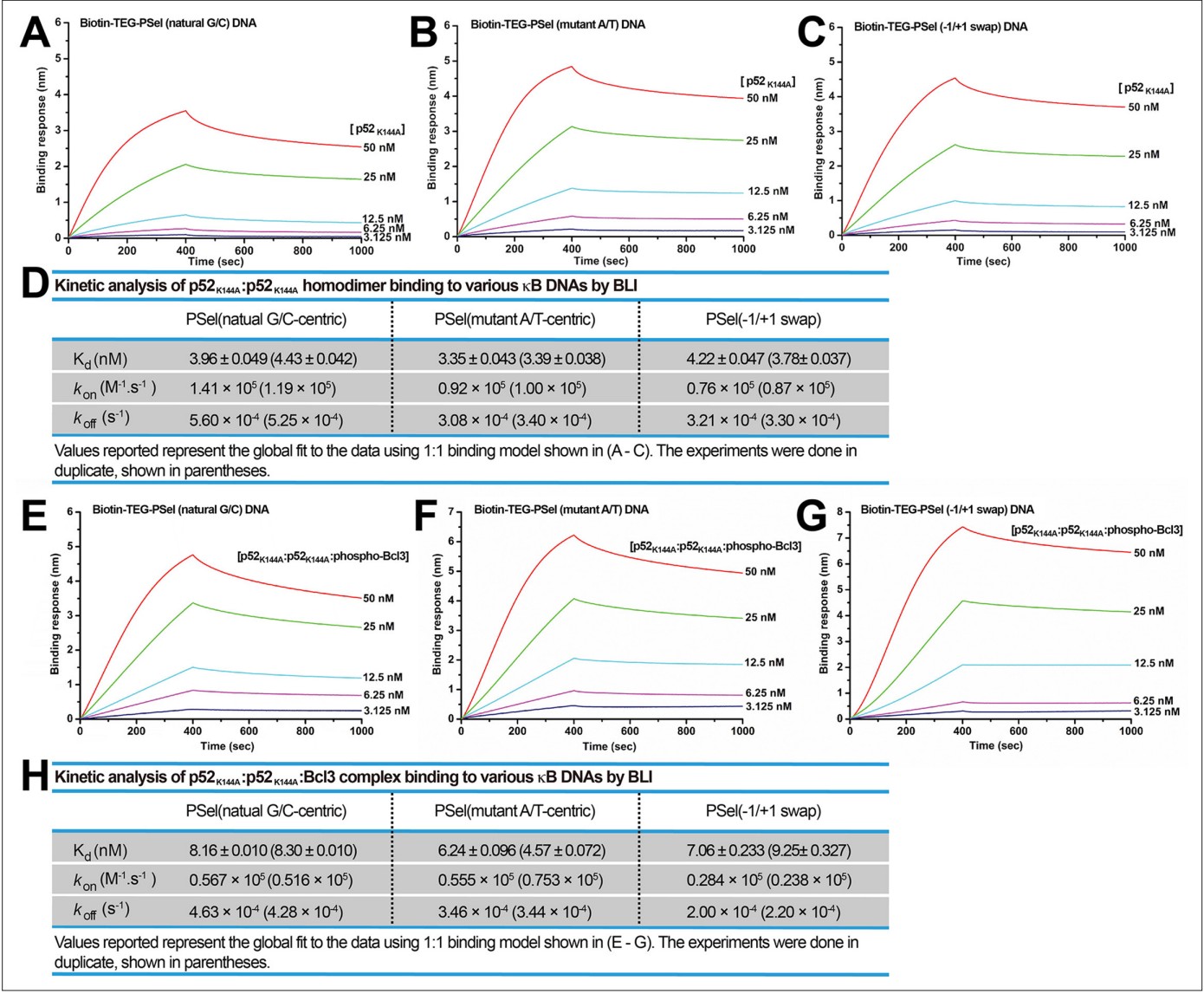

**Figure 8.** p52$_{K144A}$ binds $\kappa$ B DNAs with slower kinetics. (**A–C**) BLI binding analysis of p52$_{K144A}$:p52$_{K144A}$ (aa 1–398) homodimer to immobilized biotin labeled (**A**) natural G/C-centric, (**B**) mutant A/T-centric, and (**C**) −1/+1 swap PSel-$\kappa$ B DNAs. Each experiment was done in duplicate and one representative set of curves is shown. (**D**) Table showing the kinetic analysis in (**A–C**). (**E–G**) BLI binding analysis of p52$_{K144A}$:p52$_{K144A}$:Bcl3 complex to immobilized biotin labeled (**E**) natural G/C-centric, (**F**) mutant A/T-centric, and (**G**) −1/+1 swap PSel-$\kappa$ B DNAs. Each experiment was done in duplicate and one representative set of curves is shown. (**H**) Table showing the kinetic analysis in (**E–G**). BLI, biolayer interferometry.

The online version of this article includes the following source data and figure supplement(s) for figure 8:

**Figure supplement 1.** Recombinant p52$_{K144A}$ mutant.

**Figure supplement 1—source data 1.** Raw image of SDS-PAGE gel in *Figure 8—figure supplement 1A*, with label.

**Figure supplement 1—source data 2.** Raw image of SDS-PAGE gel in *Figure 8—figure supplement 1A*, without label.

**Figure supplement 1—source data 3.** Raw image of SDS-PAGE gel in *Figure 8—figure supplement 1D*, with label.

**Figure supplement 1—source data 4.** Raw image of SDS-PAGE gel in *Figure 8—figure supplement 1D*, without label.

To further probe the role of Lys144 identified in the simulations, this residue was mutated to Ala (p52$_{K144A}$) and assessed for binding to all three PSel-κB DNA variants. The binding kinetics of p52$_{K144A}$:p52$_{K144A}$ homodimer alone with DNAs as well as the transcriptionally competent p52$_{K144A}$:p-52$_{K144A}$:Bcl3 complex were tested. The p52$_{K144A}$ mutant was expressed and purified to similar purity as the WT p52 protein, and the mutation does not affect its interaction with Bcl3 (*Figure 8—figure supplement 1A–C*). Overall, the p52$_{K144A}$ mutant binds to all three PSel-κB DNAs with weaker affinity

compared to the WT p52 (*Figure 8*; *Figure 6*), likely due to the completely abolished Lys144 cross-strand contacts. Consistent with the simulation analysis, dissociation of all three DNAs from p52$_{K144A}$ mutant is sped up relative to the WT p52, with the most prominent change recorded for −1/+1 swap. Meanwhile, the reduction of their $k_{on}$ suggests a promotion effect of Lys144 on the closing and DNA-binding of WT p52:p52 homodimer, which may arise from the favorable electrostatic interactions between this residue and the DNAs. Compared to the natural G/C-centric DNA, the −1/+1 swap DNA binds to p52$_{K144A}$ mutant with significantly slower $k_{on}$ and $k_{off}$, and the difference between the two DNAs increases upon K144A mutation; the rate constants are in the middle of the aforementioned two DNAs in the mutant A/T-centric DNA. Overall, the substitution of the positively charged Lys144 by an alanine appears to slow down the recognition of the DNA minor grooves, which may in turn slow down the closing of p52$_{K144A}$:p52$_{K144A}$ homodimer, thereby, decreasing the association rate of the DNAs especially for the ones with narrower minor grooves such as the −1/+1 swap DNA.

## Discussion

### The p52 homodimer recognizes κB DNA using a mode distinct from other NF-κB dimers

Double-stranded DNA helices are not static entities that simply present themselves to proteins and assemble into multiprotein complexes at specific sequences. The DNA duplex is intrinsically dynamic on many levels and time scales in cells. The movement of DNA through its different conformational states is continuous and is influenced by, but not completely dependent upon, its nucleotide sequence. Structures presented in this study show that the conformations of all three PSel-κB DNA variants bound to the long p52:p52 homodimer are similar but are distinct from all previously known complexes between κB DNAs and six other NF-κB dimers (p50:p50, p50:RelA, p50:RelB, RelA:RelA, c-Rel:c-Rel, and p52:RelB). It was noted earlier a compressed minor groove in the central region of the DNA is a key feature of NF-κB-DNA complexes. The minor groove at the central three positions is significantly widened in all three complexes presented here. However, MD simulations show free DNAs exist in distinct preferred conformations, which appear to be adjusted by p52 into a unique shape for recognition. And notably, the more transcriptionally active natural PSel-κB DNA appears to maintain similar conformational and dynamic states in free and bound forms.

The current structures also demonstrate a correlation between the p52 protein length and the conformation of the κB DNA. The GRR region of the p52 protein, which was not included in any previous NF-κB structural studies, seems to play an important role. However, no electron density was observed for the GRR region in the current structures. Future studies are needed to fully understand the role of the GRR region in (p52:p52)-DNA complex conformation and the interaction with cofactor Bcl3.

### Binding affinity does not fully capture the transcriptional activity

To determine if binding affinity is related to transcriptional activity, we measured the affinity of all complexes under equilibrium condition. Surprisingly, but consistent with our previous report (*Mulero et al., 2018*), we found no correlation between affinity and transcriptional activity. The p52:p52 homodimer binds to MHC-κB with the highest affinity but it is not a transcriptional activation competent complex. Interestingly, our analysis reveals that p52:p52 homodimer uses different paths to bind κB DNAs ranging from a more entropic favored for the natural PSel, to exclusively enthalpic for MHC, and to mixed entropic-enthalpic for mutant A/T-centric and −1/+1 swap PSel DNAs. The entropy-favored processes are linked to faster binding kinetics: p52:p52 homodimer binds to natural G/C-centric PSel DNA with both faster association and dissociation rates. The most populated conformation of the free G/C-centric PSel DNA as revealed by MD simulation is similar to the one observed in the crystal structure of the complex suggesting this DNA's conformation does not undergo significant changes upon protein binding. Thus, in the complex between p52:p52 and natural G/C-centric DNA, both the DNA and protein most likely preserve their native states. This could account for the positive entropy and faster $k_{on}$. However, possibly the protein-DNA contacts in such a complex are suboptimal resulting in their faster dissociation. In contrast, p52:p52 complexes with the mutant A/T-centric or−1/+1 swap DNAs likely involve rigidification of protein-DNA contacts, requiring some structural reorganizations in both molecules,

resulting in more enthalpically stable complexes and slower association and dissociation rates. Indeed, MD simulations of the (p52:p52)-DNA complexes are consistent with this notion. The complex of p52:p52 homodimer with the natural G/C-centric PSel-κB DNA maintains very similar conformations under simulations as seen in the crystal structure. In contrast, induced by their narrowed minor groove geometries, the p52 protein closes on the DNAs in the PSel (−1/+1 swap) complex and to a lesser extent in the (mutant-A/T-centric) complex, which is likely unfavored by the p52:p52 homodimer conformation. These structural changes led to the cross-strand contacts by Lys144 in the −1/+1 swap and mutant A/T-centric DNAs but not in the natural G/C-centric PSel DNA. Combination of MD simulations and structural studies supports the model as PSel (mutant A/T-centric) and (−1/+1 swap) DNAs undergo conformational changes from free to bound states. Our observations are consistent with other studies which have shown that rapid association and dissociation are favored by entropy, whereas slow association and dissociation are guided by enthalpy (*Baerga-Ortiz et al., 2004*).

## Ideas and speculation: transcriptional regulation via kinetic discrimination

Understanding transcriptional regulation has attracted many researchers since the discovery of the *lac* operon. One of the most intriguing questions scientists are working to resolve is the mechanism of transcriptional regulation by the specific DNA response elements. Affinity regulation by different target DNA sequences for a TF has long been thought to be the dominant mode of regulation imposed by such DNA sequences. Indeed, in many cases of eukaryotic transcription differential affinity has been shown to be critical (*Sekiya et al., 2009*). TFs are also known to bind free DNA or nucleosome with distinct kinetics (*Donovan et al., 2019*). But none of these studies has established a direct correlation between binding kinetics and transcriptional regulation in eukaryotes. Many biological systems have been studied in detail with the roles of binding kinetics in regulation evaluated. For instance, receptor-ligand interactions, T cell activation, and potency of bacterial toxins are guided by the half-life of key complexes (*Corzo, 2006*; *González et al., 2005*; *Gross and Lodish, 2006*; *Nakajima et al., 2001*).

Of all six κB DNAs tested, the natural G/C-centric PSel and Skp2 DNAs showed greater transcriptional activation. We found that a slower dissociation rate or longer residence time is linked to reduced transcriptional activation. Although the relationship between the dissociation rates and transcription activities is shown for only six tested binding sites, this relationship is preserved for (p52:p52)-DNA complexes and to a lesser extent for (p52:p52:Bcl3)-DNA complexes. The relationship between binding kinetics and conformations of the complexes is further supported by the study of a mutant. This mutant, p52$_{K144A}$, binds all DNAs with a slower kinetics but the kinetics is the most prominently slowed down for the PSel (−1/+1 swap) DNA.

The rate constants obtained in our in vitro assays probably are not the same in vivo, where many other factors will have an impact on binding kinetics. However, the relative rates clearly suggest that the persistent presence of p52 on DNA gives rise to less transcription. Work presented here hints at a link between the DNA binding kinetics of a TF and its interaction with coactivators and corepressors. We previously showed that the p52:p52:Bcl3 complex preferentially recruits HDAC3 when it remains bound to an A/T-centric κB site (*Wang et al., 2012*), it is possible that the slower dissociation rate or longer residence time of p52:p52:Bcl3 on the A/T-centric κB site described in this study matches the slower on rate of the corepressor to p52:p52:Bcl3 bound to A/T DNA. That is, the A/T-centric κB DNA:p52:p52:Bcl3 complex remains stable for long enough to give the HDAC3 corepressor complex enough time to stably interact with it. In addition, the binding of other TFs to the promoters/enhancers of target genes inevitably impacts on the coactivator/corepressor regulation by the (p52:p52)-DNA complexes. Future experiments aimed at coactivator and corepressor interaction rates within the context of chromatinized DNA are needed to verify the validity of the kinetic model for DNA element sequence-specific gene regulation.

In summary, our studies have revealed a novel conformation for κB DNA in complex with NF-κB and a new organization of an NF-κB dimer. More importantly, our work provides a new insight into the mechanism of differential thermodynamics and kinetics of NF-κB-DNA binding. DNA response elements with only 1 or 2 bp variations could provoke drastically different kinetic and thermodynamic effects. Future experiments will help us fully understand how such binding processes result in transcription activation or repression.

## Materials and methods

### Protein expression and purification

Recombinant non-tagged human p52 (aa 1–398), (aa 1–398) K144A mutant, and (aa 1–327) was expressed and purified from *E. coli* Rosetta (DE3) cells. Rosetta (DE3) cells transformed with pET-11a-p52 (aa 1–398), (aa 1–398) K144A, or (aa 1–327) were cultured in 2 L of LB medium containing 50 mg/mL ampicillin and 34 mg/mL chloramphenicol at 37°C. Expression was induced with 0.2 mM Isopropyl β-D-1-thiogalactopyranoside (IPTG) at $OD_{600}$ 0.5–0.6 for 3 hr. Cells were harvested by centrifugation, suspended in 40 mM Tris-HCl (pH 7.5), 100 mM NaCl, 10 mM β-Mercaptoethanol (β-ME), 1 mM PMSF, and lysed by sonication. Cell debris was removed by centrifugation (20,000×*g* for 30 min). Clarified supernatant was loaded onto Q-Sepharose FF column (GE Healthcare). Flow through fraction was applied to SP HP column (GE Healthcare). The column was washed with 40 mM Tris-HCl (pH 7.5), 200 mM NaCl, 10 mM β-ME, and the protein was eluted by the same buffer containing 400 mM NaCl. p52 was concentrated and loaded onto the gel filtration column (HiLoad 16/600 Superdex 200 pg, GE Healthcare) pre-equilibrated with 10 mM Tris-HCl (pH 7.5), 100 mM NaCl, and 5 mM β-ME. Peak fractions were concentrated to ~10 mg/mL, flash frozen in liquid nitrogen, and stored at –80°C.

His-Bcl3 (1-446) WT and phospho-mimetic mutant was expressed in *E. coli* Rosetta (DE3) cells by induction with 0.2 mM IPTG at $OD_{600}$ 0.4 for 8 hr at 24°C. Cell pellets of 2 L culture of Bcl3 alone or together with 1 L culture of p52 (for p52:p52:Bcl3 complex) were resuspended together in buffer containing 20 mM Tris-HCl (pH 8.0), 300 mM NaCl, 25 mM imidazole, 10% glycerol, 10 mM β-ME, 0.1 mM PMSF, and 50 µL protease inhibitor cocktail (Sigma-Aldrich) and then purified by Ni Sepharose (HisTrap HP, GE Healthcare) followed by anion exchange column (Q Sepharose fast flow, GE Healthcare). The protein complex further went through HiTrap Desalting Column (GE Healthcare) to exchange buffer before BLI assays.

### Crystallization, data collection, and structure determination

Annealed DNA duplex was mixed in 20% molar excess with the pure protein.

The crystals of the p52 (aa 1–398):PSel(G/C-centric) 18 bp complex were obtained by the sitting-drop vapor diffusion method at 20°C with a reservoir solution containing 0.1 M sodium malonate (pH 4.0), 0.2 M CsCl, and 5% (w/v) PEG 3350.

The crystals of the p52 (aa 1–398):PSel(A/T-centric) 18 bp complex and the p52(aa 1–398):PSel(−1/+1 swap) 18 bp complex were obtained by the sitting-drop vapor diffusion method at 20°C with a reservoir solution containing 0.1 M sodium malonate (pH 4.0), 50 mM CsCl, and 2.5% (w/v) PEG 3350.

The crystals of the p52 (aa 1–327):Psel(A/T-centric) 13 bp complex were obtained by the sitting-drop vapor diffusion method at 20°C with a reservoir solution containing 50 mM MES (pH 6.0), 10 mM $MgCl_2$, and 10% (w/v) PEG 3350.

Before data collection, all crystals were briefly soaked in their original crystallization solution with 20% (v/v) ethylene glycol. All crystals were flash frozen in liquid nitrogen for diffraction screening and data collection at 100K. X-ray diffraction data were collected at beamline BL19U1 at Shanghai Synchrotron Radiation Facility. The initial solution was obtained by molecular replacement using Phaser (RRID:SCR_014219) (*McCoy et al., 2007*) with p52-MHC DNA complex (*Cramer et al., 1997*) as the search model. The structure was further refined through an iterative combination of refinement with Refmac5 (RRID:SCR_014225) (*Murshudov et al., 2011*) and manual building in the Coot program (RRID:SCR_014222) (*Emsley and Cowtan, 2004*; *Emsley et al., 2010*). The crystallographic information is summarized in *Table 1*.

### MD simulation

All free DNA MD simulations were carried out in GROMACS 2020.6 (*Abraham et al., 2015*) with Amber14sb force field (*Maier et al., 2015*) and OL15 parameters for DNA (*Zgarbová et al., 2015*), whereas all (p52:p52)-DNA complex MD simulations were conducted in GROMACS 2021.4 with Amber19sb force field (*Tian et al., 2020*) and OL15 parameters for DNA. To prepare the free DNA systems, crystal structures of κB/κB-like DNAs were extracted from the corresponding experimentally resolved p52:p52 dimer-bound structures, whereas the MHC DNA was retrieved from RCSB PDB database (RRID:SCR_012820) (PDB 1A3Q) (*Cramer et al., 1997*). In each system, κB DNA or (p52:p52)-DNA complex was placed in the center of a dodecahedron box with a 12 Å margin, solvated with TIP3P water (*Jorgensen et al., 1983*), and ionized with 0.1 M NaCl. Energy minimization was

performed until the maximum force of system was below 1000 kJ·mol⁻¹·nm⁻¹. The minimized system was then equilibrated in a NVT ensemble for two 1-ns stages, where the DNA heavy atoms were harmonically restrained with a force constant of 20,000 and 10,000 kJ·mol⁻¹·nm⁻², respectively. Subsequently, the system was subjected to a 6-ns position-restrained NPT equilibration, with the force constant of DNA restraint gradually reduced from 10,000 to 400 kJ·mol⁻¹·nm⁻². At the meantime, the protein heavy atoms were harmonically restrained with a force constant of 1000 kJ·mol⁻¹·nm⁻² throughout the NVT and NPT equilibrations. Finally, five replicas of 2-μs unrestrained production simulations were performed for each DNA system, resulting in an aggregated 10-μs trajectory for each free DNA system and 12-μs trajectory for each (p52:p52)-DNA complex system, with a total simulation time of 76 μs. In all simulations, van der Waals forces were smoothly switched to 0 from 9 to 10 Å. Electrostatics were calculated using the particle mesh Ewald (PME) method (*Darden et al., 1993*) with a cutoff of 10 Å. A velocity-rescaling thermostat (*Bussi et al., 2007*) was employed for the temperature coupling at 300K, whereas pressure coupling at 1 bar was implemented by a Berendsen barostat (*Berendsen et al., 1984*). All bonds involving H atoms were constrained using the LINCS algorithm (*Hess et al., 1997*).

Occupancy of DNA and p52:p52 homodimer were calculated over the corresponding integrated trajectories using the VolMap tool in visual molecular dynamics (VMD) (RRID:SCR_001820) (*Humphrey et al., 1996*). The clustering analyses were conducted within GROMACS packages (RRID:SCR_014565) using GROMOS method. Representative structures or conformations were obtained from the centroid structures of top clusters from the clustering analysis of the corresponding structural elements and rendered with PyMOL (RRID:SCR_000305) (*Schrodinger, 2015*). The hydrogen bonds were calculated using PyMOL with default standard (heavy atom distance cutoff of 3.6 Å and angle cutoff of 63°). The bp and groove parameters were measured via Curves+ (RRID:SCR_023268) (*Lavery et al., 2009*; *Blanchet et al., 2011*), with the uncertainty represented by the standard error of the mean computed from the five replica simulations of a given system.

## ITC assays

ITC measurements were carried out on a MicroCal iTC200 (Malvern Inc) at 25°C. The ITC protein sample p52 (1–398) went through desalting column (HiTrap desalting, GE Healthcare) to freshly made ITC buffer containing 20 mM Tris-HCl (pH 8.0), 100 mM NaCl, and 1 mM dithiothreitol (DTT). DNA oligos were dissolved in the same freshly made ITC buffer; equal amount of top and bottom strands of the oligos were mixed followed by heating at 95°C for 10 min and then slowly cooling down to room temperature for annealing. About 35 μM p52 (1–398) protein (in cell) was titrated with 300 μM DNAs (in syringe). A time interval of 150 sbetween injections was used to ensure that the titration peak returned to the baseline. The titration data were analyzed using the program Origin7.0 (RRID:SCR_014212) and fitted by the One Set of Site model.

## BLI assays

The kinetic assays were performed on Octet K2 (ForteBio) instrument at 20°C with shaking at 1000 RPM. The SA biosensors were used for protein-DNA interactions and were hydrated in BLI buffer containing 20 mM Tris-HCl (pH 8.0), 100 mM NaCl, 1 mM DTT, and 0.02% (v/v) Tween-20. All DNAs used were 20-mer in length and biotin-triethyleneglycol (TEG) labeled. The DNAs were loaded at 50 nM for 300 s prior to baseline equilibration for 60 s in the BLI buffer. Association of p52:p52 (aa 1–398) or p52:p52:Bcl3 complex in BLI buffer at various concentrations was carried out for 400 s prior to dissociation for 600 s. The Ni-NTA biosensor was used for protein-protein interactions and was hydrated in BLI buffer containing 20 mM Tris-HCl (pH 8.0), 200 mM NaCl, 5% glycerol, 1 mM DTT, and 0.02% (v/v) Tween-20. His-tagged-Bcl3 was loaded at 500 μg/mL for 90 s prior to baseline equilibration for 180 s in the BLI buffer. The association of p52 in BLI buffer at various concentrations was carried out for 240 s prior to dissociation for 360 s. All data were baseline subtracted and analyzed Sartorius Octet BLI analysis system (RRID:SCR_023267) using a global fitting to a 1:1 binding model. The experiments were done in duplicate.

## Luciferase reporter assays

HeLa cells were obtained from ATCC (CCL-2) (RRID:CVCL_0030). Cell cultures were tested negative for mycoplasma infection (Universal Mycoplasma Detection kit 30–1012K, ATCC). HeLa cells

were cultured in Dulbecco's modified Eagle's medium (DMEM; Gibco, catalog #11995065) that was supplemented with 10% fetal bovine serum (FBS; Gibco, catalog #10270106) and 1× Penicillin-Streptomycin-L-Glutamine (Corning, catalog #30-009Cl). HeLa cells were transiently transfected with Flag-p52 (1–415) together with Flag-Bcl3 (1–446) expression vectors or empty Flag-vector, and the luciferase reporter DNA with specific κB DNA promoter (*Wang et al., 2012*). The total amount of plasmid DNA was kept constant for all assays. Transient transfections were carried out using Lipofectamine 2000 (Invitrogen). Cells were collected 48 hr after transfection. Luciferase activity assays were performed using Dual-Luciferase Reporter Assay System (Promega) following the manufacturer's protocol. Data are represented as mean standard deviations (SDs) of three independent experiments in triplicates.

## Acknowledgements

The authors thank the Proteomics, Metabolomics and Drug Development (PMDD) Core and Prof. Qi Zhao at Faculty of Health Sciences for providing the ITC machine for the thermodynamic assays and the BLI machine for the binding assays, respectively. The authors thank the staffs from BL19U1 beamline of National Facility for Protein Science in Shanghai (NFPS) at Shanghai Synchrotron Radiation Facility, for assistance during data collection. The authors thank Prof. Liang Tong at Columbia University for critical discussion of the manuscript. This work was supported by the Science and Technology Development Fund, Macao SAR (FDCT) (project 0104/2019 /A2 and 0089/2022/AFJ to VY-FW); the Multi-Year Research Grant from University of Macau (MYRG2018-00093-FHS to VY-FW); and the computing resources of the X-GPU cluster supported by the Hong Kong Research Grant Council Collaborative Research Fund (C6021-19EF to YW). TL and YW were supported by direct grants from the Chinese University of Hong Kong. GG were supported by the National Institutes of Health (NIH) (GM085490 and CA142642 to GG).

## Additional information

### Funding

| Funder | Grant reference number | Author |
|---|---|---|
| Science and Technology Development Fund | 0104/2019/A2 | Vivien Ya-Fan Wang |
| University of Macau | Multi Year Research Grant MYRG2018-00093-FHS | Vivien Ya-Fan Wang |
| Hong Kong Research Grant Council Collaborative Research Fund | C6021-19EF | Yi Wang |
| Chinese University of Hong Kong | | Tianjie Li Yi Wang |
| National Institutes of Health | | Gourisankar Ghosh |
| Science and Technology Development Fund | 0089/2022/AFJ | Vivien Ya-Fan Wang |
| Macao SAR (FDCT) | 0104/2019/A2 | Vivien Ya-Fan Wang |
| Macao SAR (FDCT) | 0089/2022/AFJ | Vivien Ya-Fan Wang |
| National Institutes of Health | GM085490 | Gourisankar Ghosh |
| National Institutes of Health | CA142642 | Gourisankar Ghosh |

The funders had no role in study design, data collection and interpretation, or the decision to submit the work for publication.

## Author contributions
Wenfei Pan, Data curation, Formal analysis, Methodology; Vladimir A Meshcheryakov, Data curation, Formal analysis; Tianjie Li, Data curation, Formal analysis, Writing - original draft, Writing - review and editing; Yi Wang, Funding acquisition, Writing - review and editing; Gourisankar Ghosh, Conceptualization, Supervision, Writing - review and editing; Vivien Ya-Fan Wang, Conceptualization, Formal analysis, Supervision, Funding acquisition, Investigation, Writing - original draft, Project administration, Writing - review and editing

## Author ORCIDs
Tianjie Li http://orcid.org/0000-0003-4734-1577
Gourisankar Ghosh http://orcid.org/0000-0001-6311-7351
Vivien Ya-Fan Wang http://orcid.org/0000-0002-1984-2713

## Decision letter and Author response
Decision letter https://doi.org/10.7554/eLife.86258.sa1
Author response https://doi.org/10.7554/eLife.86258.sa2

# Additional files

## Supplementary files
• Supplementary file 1. Nucleotide sequences of PSel-κB DNA variants used in crystallization.
• Supplementary file 2. Assembly analysis of p52-DNA complexes made by PDBePISA (*Krissinel and Henrick, 2007*).
• Supplementary file 3. Summary of protein-DNA contacts.
• MDAR checklist

## Data availability
The atomic coordinates have been deposited in the Protein Data Bank, https://www.wwpdb.org/ (PDB ID codes 7CLI, 7VUQ, 7VUP and 7W7L).

The following datasets were generated:

| Author(s) | Year | Dataset title | Dataset URL | Database and Identifier |
| --- | --- | --- | --- | --- |
| Meshcheryakov VA, Wang VY-F | 2023 | Structure of NF-kB p52 homodimer bound to P-Selectin kB DNA fragment | https://www.rcsb.org/structure/7CLI | RCSB Protein Data Bank, 7CLI |
| Meshcheryakov VA, Wang VY-F | 2023 | Structure of NF-kB p52 homodimer bound to A/T-centric P-Selectin kB DNA fragment | https://www.rcsb.org/structure/7VUQ | RCSB Protein Data Bank, 7VUQ |
| Meshcheryakov VA, Wang VY-F | 2023 | Structure of NF-kB p52 homodimer bound to +1/-1 swap P-Selectin kB DNA fragment | https://www.rcsb.org/structure/7VUP | RCSB Protein Data Bank, 7VUP |
| Meshcheryakov VA, Wang VY-F | 2023 | Structure of NF-kB p52 homodimer bound to 13-mer A/T-centric P-Selectin kB DNA fragment | https://www.rcsb.org/structure/7W7L | RCSB Protein Data Bank, 7W7L |

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
