## [Editor Report]

This manuscript provides an important structural and biophysical characterization of several complexes of the p52 homodimer of NF kB and different DNA binding sites. The main topic is the investigation of why the central base pair(s) have a strong influence on the transcriptional activity of the homodimer. The authors correlate structural changes with measurements of kinetic on and off rates to develop a model that explains the differences in activity and supports their interpretation with MD simulations, however, some technical issues regarding differences in the ITC and BLI measurements remain unresolved. The paper will be of interest to scientists working on understanding transcriptional regulation.

---

## [Decision Letter]

**Decision letter after peer review:**

[Editors’ note: the authors submitted for reconsideration following the decision after peer review. What follows is the decision letter after the first round of review.]

Thank you for submitting your article "Structures of NF-κB p52 homodimer-DNA complexes rationalize binding mechanisms and transcription activation" for consideration by *eLife*. Your article has been reviewed by 3 peer reviewers, including Volker Dötsch as the Reviewing Editor and Reviewer #3, and the evaluation has been overseen by a Reviewing Editor and Volker Dötsch as the Senior Editor.

Essential revisions:

1) The resolution obtained for the crystal structures does not allow the identification of individual hydrogen bonds and other details. These limitations should be discussed since these details are important for the structural interpretation.

2) The data shown in Figure 1 indicate a wider minor groove for p52 homodimers with C-terminal extensions bound to long (18bp) kB sites (PSel, PSelG->A, pSelAT) and a narrower minor groove for minimal p52 homodimers bound to 13 bp kB sites (pSelAT, MHC). The authors suggest that the protein length may be important, but they do not discuss that the DNA length may also influence the structures. The same kB sequence (PSelAT) appears in both compressed and expanded minor groove forms. Analysis of the structures indicates that contacts (Ser61-A6,A7) are made outside of the 11-bp kB site in the 18-bp structures and would not be present in the 13-bp structures. So, the conclusion that p52 bound to PSel kB has a widened minor groove relative to other complexes may be correct, but it is far from solid given the data that is presented.

3) The main question of the manuscript is what is different about p52-PSel kB contacts compared to kB sites with a central A/T (like MHC)? The overall goal is to understand how a single bp can make such a large difference in the transcriptional outcome, but the answer is not clear. Loss of a cross-strand interaction is discussed, but a new Arg-Gua interaction is gained and that is not discussed. There is a lot of detail about the asymmetric nature of the complex, but most of that appears to be common to all kB sites. A better definition and discussion of the structural source of the observed selectivity is necessary.

4) It is not obvious how the raw ITC data are converted into the integrated data shown below the raw data. In principle, the dilution heat has to be subtracted followed by the integration of the peak. In the graph with the integrated data, the first 7 points in A have the same value (as is expected for such a strong binding). But the peaks in the raw data diagram vary substantially. That could be due to differences in the width of the peaks, but that should be explained as why it happens and shown. It would also help to have at least one repetition of the experiments for validation purposes.

5) Related: How was the DNA prepared for the ITC measurements? In two of the four diagrams showing the raw data substantial dilution heat of the DNA is measured. Since the DNA sequence is very similar in all cases, a similar dilution heat is expected. Is there any impurity in two of the four preparations (that would also influence the overall results)?

6) Adding an error estimate for ΔS would allow readers to evaluate whether the differences are significant. There is an amazing 3-fold variation in ΔH, from -5 kcal/mol to -15 kcal/mol. Equally amazing is that the entropic contribution TΔS also varies widely, from +5 kcal/mol to -5 kcal/mol. I found it surprising that there was no attempt to relate these findings to the known structures. Do the number of polar interactions differ? Solvent accessible surface? Amount of ordered solvent?

7) The Kd values determined from the kinetic measurements differ by > 50-fold from the calorimetry results. This discrepancy needs to be addressed. Can the on and off-rates be trusted in these experiments? Even if they are relative rates, one might think that their ratios (K) would still be relevant.

8) None of the binding events are 'entropy driven'. The most favorable (PSel kB) is still 50:50 enthalpy:entropy.

9) Why would binding with native conformations for p52 and PSel kB account for favorable entropy?

10) The authors include Bcl3 in a set of binding kinetics experiments and get similar relative rates as with p52 alone. However, the idea that Bcl3 could have an influence on site-specific transcriptional activity seems to have been ruled out because Bcl3 doesn't contact the kB site. What about an indirect mechanism? The fact that unmodified Bcl3 prevents p52 from binding DNA, yet phospo-Bcl3 does not is very suggestive of some level of involvement.

*Reviewer #1 (Recommendations for the authors):*

Meshcheryakov et al. present a compelling story showing that upon binding to DNA, an NF-κB homodimer induces distinct conformations in DNA, regardless of the DNA sequence or their level of transcriptional activity. To understand how the protein achieves transcriptional selectivity for distinct sequences, authors use simulations, binding assays, and kinetic studies. They observe a correlation between kinetics and transcriptional activation, showing that binding to active DNA sequences is driven by entropy and occurs without a large deformation of the DNA. Conversely, binding to DNA sequences with lower transcriptional activity requires more significant alteration of DNA conformation and is largely enthalpic with slower kinetics. These studies uncover an important role of binding kinetics in DNA selectivity by transcription factors that should motivate many future studies to further explore and elucidate these mechanisms.

I found this study very compelling and believe that this manuscript represents an important contribution to our understanding of DNA selectivity in TFs. To my knowledge, the role of binding kinetics has not been appreciated or investigated thoroughly and needs to be considered more, which will be inspired by this work. While I do not have any concerns that should prevent or even delay publication, I have provided a few suggestions below that I believe may enhance readability and should only improve an already strong piece of work.

Lines 76-80: authors should consider just writing out the kB consensus sequences (5'-GGGNNNNNCC-3') and explaining the numbering separately, as the current form is hard to read/parse.

Authors should include the full name of NF-κB, I don't believe I found any mention of it throughout the manuscript.

I think a few things could help to enhance the introduction section. As written, the final paragraph of this section only vaguely hints at but does not highlight the major results of the paper. Including the full sequence of psel kB, mentioning which promoters are active vs inactive, and how the MD and biochemical data help to distinguish between these where structures do not would help to strengthen this section. Basically, more specific details of the results in this section could be very nice!

Related, it could be helpful for the authors to write out the full sequences of the mutant A/T and -1/+1 swap in both introduction and again in results. Yes, readers can infer what each permutation means, but it would be nice to have it explicitly written out to avoid this step.

Line 169 is the first mention of LPS. Please define this.

This is a minor point, but could authors color figure 2D by domain? Even a lighter/darker version of the selected colors would help readers to more easily decipher the interdomain interface and make the figure clearer.

A set of interactions is described in lines 262-266 which are not shown in any figure. It would be nice to either include the figure reference (if there is one) or create a figure. E.g. I could not find Cys57 on any of the figures.

Please clarify lines 289-294. When you say swapped, which DNAs are you referring to explicitly? This part is tricky from the reader's perspective because technically, the A-T swap and control are swapped relative to the others.

Lines 128-130: referring to work by Leung et al. It is not clear what this reference means when it says 'binding of RelA homodimer to A- and T- centric DNAs are different'. Different from what? From binding to GC-centric DNA, or different from what p52 homodimer does?

Can authors specify what atoms are used in the measurement of minor groove width? This is not clear since they show ~ 2 widths per basepair in the Figure 1 table.

lines 245-251: this part got tricky to read because of the back and forth between figures 2 and 3. If there is any way the authors can move Figure 2D to be part of Figure 3, it would help to streamline the reading of this section.

The only part of this study (for me) that seems slightly disconnected from the rest is the crystallographic data from the longer protein and DNA constructs which showed a different organization of the complex. In particular, I am skeptical of the use of the phrase 'correlation between p52 protein length and conformation of kB DNA'. Correlation is a less than ideal word choice because it implies a quantifiable relationship. It is not clear how a crystal structure conformation could be correlated with anything else unless the authors were highlighting a very specific structural feature within either the protein or DNA that changes proportionally with the length. Could authors rephrase all mentions of correlation here to say something like 'observed influence of p52 length on the conformation of kB'?

Lines 418-419: authors make mention of the most populated conformation of the free DNA, but other than examining minor groove widths, the authors have not performed an actual conformational analysis. Authors could perhaps consider performing an RMSD or RMSF analysis, something that lets us see how the structure changes over the simulation or even the bases of DNA that are most flexible or stable over time. Such an analysis would lend support to even the broader claims throughout this paper that the DNA conformation is differentially influenced by protein binding.

*Reviewer #2 (Recommendations for the authors):*

The overall conclusions of this work are: (i) p52 homodimers are able to recognize kB sites with a G at the central position by naturally fitting the wider minor groove found in those sites, (ii) binding affinity of p52 to kB sites is poorly correlated with transcriptional activity but the entropic contribution to binding is correlated, (iii) higher on and off rates for p52/kB binding are associated with more transcriptionally active kB sites. The results are very interesting and address important questions that underlie mechanisms of transcriptional regulation.

The authors provide data to support their conclusions, but in some cases, the data and interpretations are not entirely convincing. The structural data support a wider minor groove for the PSel kB site, but it is not clear how p52 recognizes this difference. The loss of cross-strand hydrogen bonds near the center of the site is noted, but the addition of a canonical Arg-Gua interaction is shown but not discussed. A higher level of asymmetry in the dimer-DNA interface is described, but it is not clear how this relates to the central 3 bp in question.

The colorimetric studies are fascinating. A range of binding enthalpies from -5 to -15 kcal/mol are reported for the four kB sites studied and a range of -5 to +5 kcal/mol for the enthalpic contribution to binding (TΔS). The more favorable entropic free energy of binding is associated with kB sites that are more transcriptionally active. It is unclear what the source of these wide variations might be; the authors do not attempt to explain the results in light of the structural models.

Binding kinetic studies are also revealing and interesting. The more transcriptionally active kB sites (e.g., PSel kB) have faster on-rates and faster off-rates. The least active kB sites (e.g., MHC) have lower on-rates and lower off-rates. The authors suggest that increased resident time of p52/Bcl3 on the promoter is associated with lower transcriptional activity; a possible reason is that longer times allow the recruitment of co-repressors. The binding affinities derived from BLI experiments are more than 50-fold lower than those measured by ITC. Assuming that the ITC values are correct, this raises questions about what aspects of the kinetic measurements are valid and which might not be.

Overall, I think this is an interesting and important study, but the connections between the structural studies and the binding experiments need to be strengthened and there are several questions/issues about the individual experiments and interpretations that need to be resolved.

I found the paper a little hard to read, partly due to writing errors and the omission of critical data. Below is a list of specific concerns, major things first, then small things.

Table 1. I cannot find this table in the composite pdf (text + Figures+ supplemental). I assume this table contains crystallographic data/results, making it impossible to evaluate the quality of the structures. I have no idea what the resolution even is for any of the structures.

Structural data. The data shown in Figure 1 indicate a wider minor groove for p52 homodimers with C-terminal extensions bound to long (18bp) kB sites (PSel, PSelG->A, pSelAT) and a narrower minor groove for minimal p52 homodimers bound to 13 bp kB sites (pSelAT, MHC). The authors suggest that the protein length may be important, but they do not discuss that the DNA length may also influence the structures. The same kB sequence (PSelAT) appears in both compressed and expanded minor groove forms. Analysis of the structures indicates that contacts (Ser61-A6,A7) are made outside of the 11-bp kB site in the 18-bp structures and would not be present in the 13-bp structures. So, the conclusion that p52 bound to PSel kB has a widened minor groove relative to other complexes may be correct, but it is far from solid given the data that is presented.

What is different about p52-PSel kB contacts compared to kB sites with a central A/T (like MHC)? This is a major question since the overall goal is to understand how a single bp can make such a large difference in the transcriptional outcome, but the answer is not clear to me. Loss of a cross-strand interaction is discussed, but a new Arg-Gua interaction is gained and that is not discussed. There is a lot of detail about the asymmetric nature of the complex, but most of that appears to be common to all kB sites.

Calorimetry. Adding an error estimate for ΔS would allow readers to evaluate whether the differences are significant. There is an amazing 3-fold variation in ΔH, from -5 kcal/mol to -15 kcal/mol. Equally amazing is that the entropic contribution TΔS also varies widely, from +5 kcal/mol to -5 kcal/mol. I found it surprising that there was no attempt to relate these findings to the known structures. Do the number of polar interactions differ? Solvent accessible surface? Amount of ordered solvent?

Kinetics. The Kd values differ by > 50-fold from the calorimetry results. This discrepancy needs to be addressed. Can the on and off-rates be trusted in these experiments? Even if they are relative rates, one might think that their ratios (K) would still be relevant.

Discussion.

1. None of the binding events are 'entropy driven'. The most favorable (PSel kB) is still 50:50 enthalpy:entropy.

2. Why would binding with native conformations for p52 and PSel kB account for favorable entropy?

3. The authors include Bcl3 in a set of binding kinetics experiments and get similar relative rates as with p52 alone. However, the idea that Bcl3 could have an influence on site-specific transcriptional activity seems to have been ruled out because Bcl3 doesn't contact the kB site. What about an indirect mechanism? The fact that unmodified Bcl3 prevents p52 from binding DNA, yet phospo-Bcl3 does not is very suggestive of some level of involvement.

*Reviewer #3 (Recommendations for the authors):*

Understanding how transcriptional activity is regulated is a very important topic. The authors provide a detailed structural and biophysical characterization of several complexes of the p52 homodimer of NF kB and different DNA binding sites. The focus is on the central base pair as well as the flanking base pairs on both sides. By x-ray crystallography the authors show that the minor grove of the central base pairs is widened with the G:C base pairs compared to the A:T base pairs. Using MD simulations of the DNA the authors show that DNA molecules can adopt different conformations and the binding of the p52 homodimer induces the least conformational changes in the naturally occurring sequence. The authors go further and correlate these conformational changes with binding affinity and with kinetic k_on_ and k_off_ measurements. Overall, there is little correlation between affinity and transcriptional activity. The only correlation they observe is between fast on and off kinetics and higher transcriptional activity. This result is surprising and does not immediately provide a mechanistic interpretation of how high transcriptional activity is achieved. The authors try to provide an explanation via binding of co-repressors but the fundamental biophysical investigations for such a model are missing at the moment.

1) I have a hard time understanding how the raw ITC data are converted into the integrated data shown below the raw data. In principle, the dilution heat has to be subtracted followed by the integration of the peak. In the graph with the integrated data, the first 7 points in A have the same value (as is expected for such a strong binding). But the peaks in the raw data diagram vary substantially.

2) How was the DNA prepared for the ITC measurements? In two of the four diagrams showing the raw data substantial dilution heat of the DNA is measured. Since the DNA sequence is very similar in all cases, a similar dilution heat is expected. Is there any impurity in two of the four preparations (that would also influence the overall results)?

3) How often were the ITC data repeated? Also, the sample contained 1 mM DTT which often provides problems in ITC measurements. TCEP would have been a better choice.

4) The kD values determined with ITC and BLI differ significantly by two orders of magnitude. This needs to be explained.

[Editors’ note: further revisions were suggested prior to acceptance, as described below.]

Thank you for resubmitting the paper entitled "Structures of NF-κB p52 homodimer-DNA complexes rationalize binding mechanisms and transcription activation" for further consideration by *eLife*. Your revised article has been evaluated by a Senior Editor and a Reviewing Editor. We are sorry to say that we have decided that this submission will not be considered further for publication by *eLife* at the moment.

We had a long discussion on your paper among the editors and reviewers involved. Everyone thinks that this is a very important topic that in principle should be further investigated and published. However, in addition to not completely satisfactory answers to some technical questions (the question of the difference in dilution heat between almost identical DNA oligos and the differences between ITC and BLI measurements are still not resolved) the main question of this paper remains unanswered. While the title of your manuscript "Structures of NF-κB p52 homodimer-DNA complexes rationalize binding mechanisms and transcription activation" suggests that the manuscript explains the functional differences between the different NF-κB oligo complexes (which would indeed be a tremendous advance in our understanding of transcriptional regulation in general as well as by NF-κB in particular) , the agreement among the reviewers and editors was that at the current stage the manuscript describes some interesting observations but without offering a convincing explanation. In your rebuttal letter you state that "We admit that even with all these structures, we still don't fully understand how a single bp change makes so much difference. We believe that a fundamental difference is in the alteration of DNA dynamics. However, rather than giving a speculative answer, we would like to wait for the results of more experiments, such as longer timescale MD simulations of these protein-DNA complexes". This is a very honest statement -- which we appreciate but is also confirms that there is no real understanding of the mechanism of this important question. During the discussion with the reviewers it became clear that we are all very enthusiastic about the project but would like to see more data that support a model that explains the observed effects. One possibility would be the addition of the longer timescale MD simulations that you mentioned if they add to the clarification of the model.

All reviewers explicitly stated that they want to be involved in the evaluation of a revised manuscript and support the submission of a manuscript including the MD data.

---

## [Author Response]

[Editors’ note: the authors resubmitted a revised version of the paper for consideration. What follows is the authors’ response to the first round of review.]

Essential revisions:1) The resolution obtained for the crystal structures does not allow the identification of individual hydrogen bonds and other details. These limitations should be discussed since these details are important for the structural interpretation.

The resolution issue has been clarified in lines 220-222: It should be noted that all the complex structures are at ~3.0 Å resolution, which places limits on the identification of some detailed interactions, including hydrogen bonds.

2) The data shown in Figure 1 indicate a wider minor groove for p52 homodimers with C-terminal extensions bound to long (18bp) kB sites (PSel, PSelG->A, pSelAT) and a narrower minor groove for minimal p52 homodimers bound to 13 bp kB sites (pSelAT, MHC). The authors suggest that the protein length may be important, but they do not discuss that the DNA length may also influence the structures. The same kB sequence (PSelAT) appears in both compressed and expanded minor groove forms. Analysis of the structures indicates that contacts (Ser61-A6,A7) are made outside of the 11-bp kB site in the 18-bp structures and would not be present in the 13-bp structures. So, the conclusion that p52 bound to PSel kB has a widened minor groove relative to other complexes may be correct, but it is far from solid given the data that is presented.

Various DNA lengths ranging from 12 to 20 bp were used in the co-crystallization, and the details have been added to Supplementary File 1. The long p52 (aa 1-398) protein crystallized with 18 and 20 bp PSel-κB DNAs and diffracted better with the 18 bp DNA. The short p52 (aa 1-327) protein only crystallized with the PSel (mutant A/T-centric) DNA at 13 bp length but not any longer DNAs; it did not crystallize with the PSel (natural G/C-centric) or (-1/+1 swap) DNAs at any length. This has been added in lines 214-217 and lines 254-257. In addition, as mentioned by the reviewer, we also mentioned in the text that “Ser61 also makes direct contact with A at ±6 and ±7 positions; these contacts are not possible for the short p52 (aa 1-327) co-crystallized with 13bp κB DNAs such as MHC and PSel (mutant A/T-centric)-κB (Figure 3—figure supplement 1A-B)” (lines 296-299).

The length and sequence of DNA also influence the structures, and we believe they are corelated with the length of p52 protein (lines 267-268). The direct contacts between the protein and A at ±6 and ±7 base pairs are a direct example of this. We did not emphasize the discussion on the ±6 and ±7 contacts because we do not fully understand the connectivity between the protein and DNA sequence and lengths.

3) The main question of the manuscript is what is different about p52-PSel kB contacts compared to kB sites with a central A/T (like MHC)? The overall goal is to understand how a single bp can make such a large difference in the transcriptional outcome, but the answer is not clear. Loss of a cross-strand interaction is discussed, but a new Arg-Gua interaction is gained and that is not discussed. There is a lot of detail about the asymmetric nature of the complex, but most of that appears to be common to all kB sites. A better definition and discussion of the structural source of the observed selectivity is necessary.

The reviewer is absolutely correct. We ask the same question and present here our best understanding based on the currently available information. We admit that even with all these structures, we still don’t fully understand how a single bp change makes so much difference. We believe that a fundamental difference is in the alteration of DNA dynamics. However, rather than giving a speculative answer, we would like to wait for the results of more experiments, such as longer timescale MD simulations of these protein-DNA complexes. To connect the difference between binding strategies to transcription is even more challenging. Nonetheless, we think continued careful structural and binding analyses will lead to an answer.

The p52 (aa 1-398) Arg52 in the complex with long 18 bp DNAs loses the cross-strand interaction; however, the NH1 and NH2 groups of Arg52 form H-bonds with both the O6 and N7 groups of G at ±3. The same Arg52 in the complex with the short 13 bp PSel (mutant A/T-centric) or the MHC-κB DNA only hydrogen bonds with the O6 group, but not the N7, of G at ±3. The homologous Arg54 in p50 also only contacts the O6 group of G at −3 in the p50:RelA-IFNb-κB DNA complex. We do not know if these contacts are of similar or different strengths. However, these variations in contacts suggest alterations in DNA dynamics (as discussed below in point #9) when the length or sequence changes. This has been added in lines 279-284.

4) It is not obvious how the raw ITC data are converted into the integrated data shown below the raw data. In principle, the dilution heat has to be subtracted followed by the integration of the peak. In the graph with the integrated data, the first 7 points in A have the same value (as is expected for such a strong binding). But the peaks in the raw data diagram vary substantially. That could be due to differences in the width of the peaks, but that should be explained as why it happens and shown. It would also help to have at least one repetition of the experiments for validation purposes.

We thank reviewer’s suggestion. The ITC experiments in Figure 5 have been repeated using a new batch of oligos and are shown in Supplemental Figure 5—figure supplement 1. In the repeated data, the MHC DNA still shows the highest affinity with p52; and the entropy and enthalpy values follow the same trend as the original data even though the exact values vary.

For the data integration, the area under the curve (gray shadow in Author response image 1) corresponds with the total heat generated, from which ΔH is determined and integrated to a dot. Both the depth and width of the first 7 peaks in panel A vary in the upper raw data diagram, resulting in similar integrated peak areas for the bottom panel. The same is true for the other panels of the figure. A detailed explanation has been added to the corresponding figure legend.

**Author response image 1. sa2fig1:** 

5) Related: How was the DNA prepared for the ITC measurements? In two of the four diagrams showing the raw data substantial dilution heat of the DNA is measured. Since the DNA sequence is very similar in all cases, a similar dilution heat is expected. Is there any impurity in two of the four preparations (that would also influence the overall results)?

The DNA oligos were dissolved in freshly made ITC buffer (same as the p52 protein). An equal amount of top and bottom strands of the oligos were mixed, followed by heating at 95°C for 10 minutes and slowly cooling down to room temperature for annealing. This has been added to the Materials and methods section for ITC (lines 626-629). The mass-spec data on DNA purity (submitted separately as Related Manuscript Files) shows that all DNA strands are highly pure, suggesting impurity is possibly not the source of the observed differences in dilution heats. Moreover, in the repeat experiments, the same two DNAs (both G/C centric, panels A and C, while the other two are A/T centric) showed larger dilution heats similar to the original experiments, even though they were from a new batch. Therefore, the different dilution heats might be an inherent property of the DNAs.

6) Adding an error estimate for ΔS would allow readers to evaluate whether the differences are significant. There is an amazing 3-fold variation in ΔH, from -5 kcal/mol to -15 kcal/mol. Equally amazing is that the entropic contribution TΔS also varies widely, from +5 kcal/mol to -5 kcal/mol. I found it surprising that there was no attempt to relate these findings to the known structures. Do the number of polar interactions differ? Solvent accessible surface? Amount of ordered solvent?

The error estimates for ΔS are not reported by the software on the ITC instrument. The ITC machine directly generates K (which is 1/Kd) and ΔH with the confidence intervals after each run. Please see Author response image 2 from the ITC machine directly as an example of the data shown in Figure 5A. From the K value (1.27x10^7^ ± 4.75x10^6^ M^-1^) in the picture, the Kd was calculated by 1/K = [1/(1.27 x10^7^ M^-1^)]x10^9^ = 78.7 nM.

We have calculated the surface area, the buried area in the p52 dimer and in the DNA interface, and the H-bonds, which are shown in the table below (and added as the new Supplementary File 2). There is no significant difference among the structures, although it is often not easy to relate thermodynamic or kinetic binding parameters to static structures which do not explain the dynamic path to binding. In addition, the solvent structure in the complexes is not clearly defined due to the limited resolution of the structures.

7) The Kd values determined from the kinetic measurements differ by > 50-fold from the calorimetry results. This discrepancy needs to be addressed. Can the on and off-rates be trusted in these experiments? Even if they are relative rates, one might think that their ratios (K) would still be relevant.

This is often an issue with the binding affinity of a macromolecular complex measured by different methods. We believe that tethering of the DNA to a solid surface in the BLI assays differentially affects its dynamics, an important component in the binding mechanism. However, the advantage of BLI is that it provides the ability to monitor the rates of association and dissociation in real-time through measuring the increase in the optical thickness at the biosensor tip. ITC, on the other hand, uses very high, non-physiological concentrations of the components. The gel-based EMSA is another common method for measuring protein-DNA/RNA affinity, which also shows different values. In that case, the complexes are running through the gel pores in low salt buffers which is far from the native binding conditions. In the end, we think the relative values are more significant than the absolute numbers. The Kd values from BLI assays follow the same trend as those from ITC.

8) None of the binding events are 'entropy driven'. The most favorable (PSel kB) is still 50:50 enthalpy:entropy.

We thank the reviewer for pointing this out. We have revised the writing in the Result section as “binding of p52 to the natural G/C-centric PSel-κB DNA is associated with a large increase in entropy (∆S) and a moderate decrease in enthalpy (∆H). On the other hand, the binding to the MHC and mutant A/T-centric PSel DNAs showed a much larger decrease in enthalpy. These results suggest that the binding of p52:p52 homodimer to the G/C-centric κB DNA is favored more by entropy, whereas the binding to the A/T-centric DNA is driven by enthalpy alone” (lines 391-398). Other places in the discussion have also been revised accordingly.

9) Why would binding with native conformations for p52 and PSel kB account for favorable entropy?

Based on the MD simulations, the most populated conformation of the free G/C-centric PSel DNA is similar to the one observed in the crystal structure of the complex, suggesting this DNA’s conformation does not undergo significant changes upon protein binding. Thus, in the complex between p52:p52 and natural G/C-centric DNA, both the DNA and protein most likely preserve their native states. This could account for the positive entropy and faster *k*_on_. However, possibly the protein-DNA contacts in such a complex are sub-optimal, resulting in their faster dissociation. In contrast, p52:p52 complexes with the mutant A/T-centric or −1/+1 swapped DNAs likely involve rigidification of protein-DNA contacts, requiring some structural reorganizations in both molecules, resulting in more enthalpically stable complexes and slower association and dissociation rates. This is included in the Discussion section, lines 484-494.

10) The authors include Bcl3 in a set of binding kinetics experiments and get similar relative rates as with p52 alone. However, the idea that Bcl3 could have an influence on site-specific transcriptional activity seems to have been ruled out because Bcl3 doesn't contact the kB site. What about an indirect mechanism? The fact that unmodified Bcl3 prevents p52 from binding DNA, yet phospo-Bcl3 does not is very suggestive of some level of involvement.

The reviewer is correct in that phosphorylation alters Bcl3’s transcriptional activity. We are currently performing experiments to understand how phospho-Bcl3 becomes a transcriptional cofactor by inducing the formation of the ternary complex. This is counterintuitive since phosphorylation of one of the three residues (S446) in the C-terminus, which possibly lies near the negatively charged DNA, is expected to be repulsive. We believe that there is a unique structural component regulating this event where the C-terminus of phospho-Bcl3 adopts a distinct conformation and moves away from the protein-DNA interface. We hope we will be able to clarify in the future how this phosphorylation-dependent conformational variation dictates Bcl3’s transcriptional activity.

[Editors’ note: what follows is the authors’ response to the second round of review.]

We had a long discussion on your paper among the editors and reviewers involved. Everyone thinks that this is a very important topic that in principle should be further investigated and published. However, in addition to not completely satisfactory answers to some technical questions (the question of the difference in dilution heat between almost identical DNA oligos and the differences between ITC and BLI measurements are still not resolved) the main question of this paper remains unanswered. While the title of your manuscript "Structures of NF-κB p52 homodimer-DNA complexes rationalize binding mechanisms and transcription activation" suggests that the manuscript explains the functional differences between the different NF-κB oligo complexes (which would indeed be a tremendous advance in our understanding of transcriptional regulation in general as well as by NF-κB in particular) , the agreement among the reviewers and editors was that at the current stage the manuscript describes some interesting observations but without offering a convincing explanation. In your rebuttal letter you state that "We admit that even with all these structures, we still don't fully understand how a single bp change makes so much difference. We believe that a fundamental difference is in the alteration of DNA dynamics. However, rather than giving a speculative answer, we would like to wait for the results of more experiments, such as longer timescale MD simulations of these protein-DNA complexes". This is a very honest statement -- which we appreciate but is also confirms that there is no real understanding of the mechanism of this important question. During the discussion with the reviewers it became clear that we are all very enthusiastic about the project but would like to see more data that support a model that explains the observed effects. One possibility would be the addition of the longer timescale MD simulations that you mentioned if they add to the clarification of the model.

We deeply appreciate that all reviewers found our project to be interesting and important to study. Based on reviewers’ suggestions, we have now further revised the manuscript. We have carried out microsecond MD simulations of the three (p52:p52)-DNA complexes, including the natural G/C-centric, mutant A/T-centric and −1/+1 swap PSel-κB DNAs. The complex of p52:p52 homodimer with the more transcriptionally active natural G/C-centric DNA maintains very similar conformations under simulations as seen in the crystal structure. In contrast, induced by their narrowed minor groove geometries, the p52 protein closes on the DNAs in the −1/+1 swap complex and to a lesser extent in the mutant A/T-centric complex, which is likely unfavored by the p52:p52 homodimer conformation. These structural changes led to the cross-strand contacts by Lys144 in the −1/+1 swap and mutant A/T-centric DNAs but not in the natural G/C-centric PSel DNA. Combination of MD simulations and structural studies supports the model as PSel (mutant A/T-centric) and (−1/+1 swap) DNAs undergo conformational changes from free to bound states. The relationship between binding kinetics and conformations of the complexes is further supported by the mutational study of this Lys144 residue. These results have been added to a new section “Differential minor groove geometries correlate with differential DNA binding kinetics” (lines 418-473), with new Figures 7-8, and Figure 7—figure supplement 1-2, Figure 8—figure supplement 1.

We believe, with the addition of the new data, our work highlights the importance of DNA sequence-dependent dynamics in protein-DNA recognitions.